Skeletal anatomy of the early Permian parareptile Delorhynchus with new information provided by neutron tomography

Rowe Dylan C. T. 1 2 dylan.rowe@mail.utoronto.ca
http://orcid.org/0000-0002-3502-4649 Bevitt Joseph J. 3
http://orcid.org/0000-0002-7454-1649 Reisz Robert R. 1 2
1 Department of Biology, University of Toronto, Mississauga , Mississauga, Ontario , Canada
2 Dinosaur Evolution Research Center, Jilin University , Changchun , China
3 Australian Centre for Neutron Scanning, Australian Nuclear Science and Technology Organization , Sydney, New South Wales , Australia
Badenhorst Shaw
Electronic publication date: 2023 Aug 22
Publication date: 2023
Volume: 11
Electronic Location ID: e15935
Received 2023 Feb 2; Accepted 2023 Jul 31
Copyright: © 2023 Rowe et al.
Copyright year: 2023
Copyright holder: Rowe et al.
License: This is an open access article distributed under the terms of the Creative Commons Attribution License, which permits unrestricted use, distribution, reproduction and adaptation in any medium and for any purpose provided that it is properly attributed. For attribution, the original author(s), title, publication source (PeerJ) and either DOI or URL of the article must be cited.
License URL: https://creativecommons.org/licenses/by/4.0/

Keywords: Paleontology, Phylogenetics, Morphology, Neutron computed tomography, Parareptilia, Anatomy, Early amniote

Funding: University of Toronto, Jilin University and a Natural Sciences and Engineering Research Council Discovery RGPIN-2020-04959 This study was made possible by funding from the University of Toronto, Jilin University and a Natural Sciences and Engineering Research Council Discovery Grant (RGPIN-2020-04959) to Robert R. Reisz. The funders had no role in study design, data collection and analysis, decision to publish, or preparation of the manuscript.

==============================
Detailed description of the holotype skeleton of Delorhynchus cifellii, made possible through the use of neutron tomography, has yielded important new information about the cranial and postcranial anatomy of this early Permian acleistorhinid parareptile. Hitherto unknown features of the skull include a sphenethmoid, paired epipterygoids and a complete neurocranium. In addition, the stapes has been exposed in three dimensions for the first time in an early parareptile. Postcranial material found in articulation with the skull in this holotype allows for the first detailed description of vertebrae, ribs, shoulder girdle and humerus of an acleistorhinid parareptile, allowing for a reevaluation of the phylogenetic relationships of this taxon with other acleistorhinids, and more broadly among parareptiles. Results show that Delorhynchus is recovered as the sister taxon of Colobomycter, and ‘acleistorhinids’ now include Lanthanosuchus.

Introduction

The reptilian clade known as Parareptilia was first erected by Olson (1947) as one of two subclasses of reptiles. In Tsuji & Müller (2009), Parareptilia was previously defined as the most inclusive clade containing Milleretta rubidgei and Procolophon trigoniceps (Tsuji & Müller, 2009). The fossil record indicated that parareptiles originated within the late Carboniferous (Modesto et al., 2015), diversified into numerous morphotypes during the Permian and persisted into the Triassic (MacDougall, Brocklehurst & Fröbisch, 2019). The body plans exhibited by members of Parareptilia in the early Permian included the bipedal herbivorous bolosaurids, highly specialized aquatic mesosaurs and insectivorous acleistorhinids (Berman et al., 2000; Modesto, 2006; Modesto, Scott & Reisz, 2009).

One of the most important locales for understanding the early stages of parareptile evolution is the Richards Spur locality in Oklahoma. Over 40 distinct tetrapod taxa have been discovered here, including nine described members of the Parareptilia (MacDougall et al., 2017a). These include the basal parareptile Microleter mckinzieorum, the bolosaurid Bolosaurus grandis, the nyctiphruretid Abyssomedon williamsi, the lanthanosuchoid Feeserpeton oklahomensis and the acleistorhinids Colobomycter pholeter, Delorhynchus priscus, D. cifellii, C. vaughni and D. multidentatus (Vaughn, 1958; Fox, 1962; Reisz, Barkas & Scott, 2002; Tsuji, Müller & Reisz, 2010; MacDougall & Reisz, 2012, 2014; Reisz, Macdougall & Modesto, 2014; MacDougall, Modesto & Reisz, 2016; Rowe et al., 2021). Early parareptiles are rare in the fossil record, and most occurrences outside of the southern United States are members of the Bolosauridae (Berman et al., 2000; Reisz et al., 2007; Müller, Li & Reisz, 2008). In the absence of other evidence, it can be stated that Richards Spur was a center for acleistorhinid diversification as almost all members of the Acleistorhinidae are found in Oklahoma. The preservation of such a high taxic diversity of parareptiles suggests that this region was a center of parareptilian diversification in the early Permian, especially for acleistorhinids (MacDougall, Modesto & Reisz, 2016). Most of the early parareptilian taxa from this locality are known from cranial material, with the exception of the holotype of D. cifellii, which was preserved with a partial vertebral column and shoulder girdle (Reisz, Macdougall & Modesto, 2014). This holotype is the first known articulated skeleton of an acleistorhinid parareptile. The genus of Delorhynchus was originally based upon a partial maxilla as D. priscus (Fox, 1962). Subsequent studies pertaining to D. cifellii include the description of a skull with preserved insect cuticle (Modesto, Scott & Reisz, 2009), as well as a ontogenetic growth series of the jugal and a thorough description of mandibular elements (Haridy et al., 2016; Haridy, Macdougall & Reisz, 2018). A third species, D. multidentatus, was recently described as the first early parareptile to possess multiple tooth rows (Rowe et al., 2021).

The specimen examined in detail within this study is the holotype of Delorhynchus cifellii, which consists of a mostly complete skull and partial postcrania. The only discernable difference between the genotype, D. priscus, and D. cifellii is the lack of a well-developed anterodorsal maxillary flange, mainly because the known specimens of D. priscus are restricted to maxillary and dentary fragments. D. cifellii differs from the third described species, D. multidentatus by the presence of a singular dentary tooth row, the absence of an enlarged dentary foramen and larger overall size. In Reisz, Macdougall & Modesto (2014), OMNH 73362, OMNH 73363 and OMNH 73524 were also used to provide other details of the cranial anatomy and dentition of D. cifellii but the holotype was not studied in detail because of the presence of extensive varanopid fossil material and matrix covering much of the skeleton. As a result of the usage of neutron computed tomography (nCT), much of the skeleton has been made available for segmentation and the current project has been able to expand significantly on the original description of the holotype.

Materials and Methods

OMNH 73515 was scanned using micro-CT at the DINGO thermal-neutron radiography/tomography/imaging station (Garbe et al., 2015) located at the 20 MW Open-Pool Australian Lightwater (OPAL) reactor housed at the Australian Nuclear Science and Technology Organization (ANSTO), Lucas Heights, New South Wales, Australia. The methodology of data acquisition follows the procedure of Mays, Bevitt & Stillwell (2017). OMNH 73515 was reconstructed with a voxel size of 45.5 × 45.5 × 45.5 µm.

The resulting scan of 2017 slices was processed in ImageJ to minimize file size and improve clarity of data, and then modeled using Avizo 3D segmentation software. All figures were compiled in Adobe Photoshop Elements 8.0. The phylogenetic analysis in this study used PAUP 4.0a149, and the matrix from Cisneros et al. (2021) was updated in Mesquite. A bootstrap analysis was conducted with 1,000 replicates of full heuristic searches, with equally weighted and unordered characters and one tree saved per step. Heuristic searches within the bootstrap analysis were performed via stepwise addition with 1,000 replicates. Where required, new information was acquired by personal examination of specimens. Bremer decal values were not used due to the weakness of the tree, and branches were collapsed if maximum branch length was zero.

Systematic paleontology

PARAREPTILIA Olson, 1947

ANKYRAMORPHA deBraga & Reisz, 1996

DELORHYNCHUS Fox, 1962

DELORHYNCHUS CIFELLII Reisz, Macdougall & Modesto, 2014

Description

General cranial and postcranial proportions

The skull of OMNH 73515 is approximately 41.2 mm long, with an orbital length of 12.8 mm resulting in a ratio of skull length to orbital length of 3.23 (Fig. 1). Mid-dorsal vertebrae are 4.38 mm long, and consequently the skull is equal to 9.4 vertebrae in length. The humerus is 27.9 mm long, the scapulae have a maximum length of 21.65 mm and a maximum height of 21.75 mm.

Figure 1 Holotype of Delorhynchus cifellii, OMNH 73515.

(A) Dorsal view and (B) ventral views. Abbreviations: sg, shoulder girdle; vc, vertebral column.

Skull roof

A slender dorsal process of the right premaxilla has been preserved contacting the adjacent nasal in OMNH 73515 (Figs. 2A and 2C). No other recognizable features of the premaxilla are present due to postmortem erosive processes, but several small fragments have been tentatively identified as part of the premaxilla.

Figure 2 Cranial material of Delorhynchus cifellii, OMNH 73515.

(A) Dorsal, (B) lateral and (C) ventral views. Abbreviations: an, angular; art, articular; bh, basihyal; bo, basioccipital; ch, ceratohyal; cr2, second coronoid ossification; d, dentary; ect, ectopterygoid; epi, epipterygoid; f, frontal; j, jugal; la, lacrimal; m, maxilla; n, nasal; nc, neurocranium; p, parietal; pal, palatine; pbs, parabasisphenoid; pf, postfrontal; po, postorbital, pp, postparietal; prf, prefrontal; pt, pterygoid; q, quadrate; qj, quadratojugal; s, stapes; sm, septomaxilla; sp, splenial; sph, sphenethmoid; sq, squamosal; st, supratemporal; v, vomer.

The right maxilla is mostly complete, extending from the ventral border of the external nares to the level of the mid orbit (Figs. 2B, 3 and 4). There are 23 tooth positions present on this maxilla, similar to that in OMNH 73362 despite its smaller size. However, the premaxillary process of the left maxilla is missing, as well as tooth-bearing positions underneath the dorsal lamina. The posterior process that underlies the left orbit has also broken off of from the rest of the maxilla, but is still closely associated with the rest of the specimen. Shallow foramina dot the anterolateral surface of the maxilla, with a deeper anterolateral maxillary foramen found at the posterior external nares which is typical of early parareptiles. This foramen extends posteromedially into the maxilla and is exposed on its medial surface at the middle of the dorsal lamina. The preserved premaxillary process is short, forming the ventral portion of the external naris. A small ridge extends from the tip of the lateral surface of the premaxillary process to the first tooth position, which would most likely form a contact with the tooth-bearing portion of the premaxilla. On the medial surface of the premaxillary process, a groove extends from the premaxillary process to the posterior orbit with increasing concavity at the anterolateral maxillary foramen. The external nares are also completely bordered posteriorly by the dorsal lamina. As with Colobomycter pholeter, C. vaughni and Acleistorhinus pteroticus (deBraga & Reisz, 1996; Modesto, 1999; MacDougall, Modesto & Reisz, 2016), an anterolateral projection of the dorsal lamina contributes to the posterodorsal external nares. In contrast, Karutia fortunata lacks prominent anterior projections of the dorsal lamina (Cisneros et al., 2021). Towards the maxilla-palatine contact, the tooth-bearing portion of the maxilla is thickened, which extends to the anterior corner of the orbit. Of the maxillary tooth positions, only eleven teeth in the right maxilla are complete, with broken or missing teeth comprising the remaining thirteen positions. None of the teeth in the left maxilla are complete, and only seventeen positions are preserved due to the missing premaxillary process. Posteriorly, the maxilla underlies with the ventral surface of the jugal as a thin terminal process but does not contact with the quadratojugal. The anatomy of the teeth have been previously described in MacDougall & Reisz (2014) as generally conical with little recurvature distally on the crown. Dentition of the maxilla is largely homodont in shape and isodont in size, with the teeth in the mid-region of the maxilla being only slightly enlarged than the rest of the rest of the tooth row. As expected, posterior maxillary teeth tend to decrease in crown height near the terminus of the bone. The tooth crowns carry delicate mesial and distal cutting edges, and the lingual surfaces have very delicate longitudinal striations.

Figure 3 Right maxilla of Delorhynchus cifellii, OMNH 73515.

(A) Lateral and (B) medial views.

Figure 4 Internal view of skull roof of Delorhynchus cifellii, OMNH 73515.

(A) Ventral view of skull roof and (B) medial view of right skull roof. Abbreviations: f, frontal; j, jugal; m, maxilla; n, nasal; p, parietal; pf, postfrontal; po, postorbital; pp, postparietal; prf, prefrontal; qj, quadratojugal; sm, septomaxilla; sq, squamosal; st, supratemporal; t, tabular.

The nasal is a paired element narrow anteriorly where it connects to the premaxilla and broader posteriorly where it connects to the maxilla and lacrimal laterally, as well as the prefrontal and frontal dorsally (Figs. 2A, 2B and 4). Its overall shape is comparable to that of Acleistorhinus pteroticus (figures 1–3 in Daly, 1969). It is possible to discern that a large anterolateral shelf that also forms the dorsomedial border of the external nares, similar to what is seen in C. pholeter (figure 3d, e in MacDougall et al., 2017b). Immediately posterior to the anterolateral shelf, a thickened region of the nasal supports the anterolateral process of the maxillary dorsal lamina. Towards the tip of the snout, a small gap in the anteromedial surface of the nasal provides a short suture for the dorsal process of the premaxilla. The posterolateral surface of the nasal contacts the anterior lateral exposure of the lacrimal. Another smooth suture between the nasal and the prefrontal extends posteromedially on the dorsal surface towards the midline of the skull. On the ventral surface of the nasal, the medial surface is smooth, only slightly concave and extends to the opening of the external nares. This smooth medial surface is bordered laterally by a shallow longitudinal ridge on each side extending into the frontal, with one set of paired foramina at the middle of the ridge. It is likely that this ridge contained the orbitonasal vein. Small, shallow foramina dot the otherwise smooth surface of the nasal.

OMNH 73515 has a partially preserved septomaxilla on the right side of the snout, whereas on the left side it has been lost due to postmortem fragmentation. What remains of the septomaxilla contacts the premaxillary process of the maxilla and the anterolateral shelf of the nasal (Figs. 2B and 4B), similar to what is present in Acleistorhinus pteroticus and Colobomycter pholeter (figure 6 in Daly, 1969; figure 4B in MacDougall et al., 2017b). It is unclear, based on the extent of preservation, whether a septomaxillary foramen would be present as in C. pholeter (figure 4B in MacDougall et al., 2017b).

As previously described in Reisz, Macdougall & Modesto (2014), the lacrimal possesses two lateral exposures on the external surface of the snout—a posterior exposure forming the anterior corner of the orbit, and a smaller anterior exposure between the maxilla and the nasal (Figs. 2B and 4B). This unusual feature, with a significant portion of the bone being covered on the external surface of the snout, was first described in Colobomycter pholeter and appears to be present in Karutia fortunata (MacDougall et al., 2017b; Cisneros et al., 2021). The double exposure noted in these acleistorhinids appears to be absent in C. vaughni where the dorsal lamina of the maxilla and the anterior part of the prefrontal overlies any anterior exposure (MacDougall, Modesto & Reisz, 2016). According to the redescription of Acleistorhinus pteroticus by deBraga & Reisz (1996), the presence of a secondary lateral exposure of the lacrimal is not noted, but that area of the skull roof was damaged and fragmented on both sides, and such a small portion of the lacrimal is difficult to identify.

The internal surface of the lacrimal is much more extensive than the laterally exposed surface, and the well-developed anterior process of the lacrimal heavily buttresses the dorsal lamina of the maxilla (Fig. 4B). This medial surface of the lacrimal appears to be a continuation of the robust region of the nasal supporting the anterolateral process of the maxillary dorsal lamina, as the anteromedial portion of the lacrimal merges with this surface of the nasal. Two pairs of lacrimal puncti are present on the thickened orbital surface of the lacrimal, a smaller dorsal punctum and a larger ventral punctum. The lacrimal canal extends from these lacrimal puncti and is bordered entirely by the posterior half of the anterior lacrimal. Anteriorly, the lacrimal canal continues into the maxilla and merges with the external nares. According to MacDougall, Modesto & Reisz (2016), the lacrimal duct is bordered by the lacrimal posteriorly and the maxilla anteriorly, but the lacrimal does not extend forward anteromedially. Laterally, the lacrimal duct is enclosed by the maxilla. Below the lacrimal puncti, the lacrimal contacts the palatine in a smooth suture. On the medial surface of the posterior lacrimal, a tall ridge extends posteromedially on each side from the anterior corner of the orbit. This antorbital wall has been previously noted to be present in procolophonids such as Procolophon and Owenetta (Carroll & Lindsay, 1985; Reisz & Scott, 2002), as well as Macroleter (Tsuji, 2006). In comparison to the aforementioned parareptiles, the antorbital wall is considerably taller and more slender in Delorhynchus cifellii. The posterior portion of the lacrimal also extends into the ventral edge of the orbit above the pronounced dorsomedial channel of the maxilla, terminating at the anterior process of the jugal.

The right jugal is complete and articulated in OMNH 73515, suturing onto the maxilla anteroventrally, the pterygoid medially and the postorbital posteriorly (Figs. 2B and 4). On the left side of the skull roof, the left jugal has been disarticulated posterolaterally by postmortem fragmentation, with the posterior edge and processes missing, but is articulated with the maxilla via its slender suborbital process. As mentioned previously, the anterior process of the jugal also has a point contact with the posterior process of the lacrimal. While the posterior process of the jugal does not contact the squamosal as noted in Haridy et al. (2016), the posterolateral skull roof is partially disarticulated and it is possible that the jugal would connect with the squamosal. The posterior process of the jugal does not contact the squamosal in this specimen, which matches what has been suggested in Haridy et al. (2016). Concave, jagged emargination of the posterodorsal process of the jugal forms the anterior border of the temporal fenestra. The latter can be divided into dorsal and ventral regions due to the posterior processes of the jugal. As previously described in Reisz, Macdougall & Modesto (2014), paired, wedge-shaped alary processes extend medially to contact the pterygoid in a fully articulated individual. Multiple shallow dimples dot the lateral surface of the jugal, which are less pronounced than in Colobomycter vaughni, C. pholeter and Acleistorhinus pteroticus (deBraga & Reisz, 1996; MacDougall, Modesto & Reisz, 2016; MacDougall et al., 2017b) but are similar in their overall pattern.

The prefrontal is a relatively large element, extending laterally from the midpoint of the maxillary dorsal lamina to the top of the orbital rim. It is ornamented externally with pronounced tuberosities and deep circular dimples (Figs. 2A, 2B and 4). Anteriorly a small, curved suture occurs between the underlying anterior plate of the prefrontal and the posterolateral end of the nasal. This anterior plate also sutures onto the medial surface of the anterior process of the lacrimal and the posterior half of the dorsal lamina of the maxilla. The antorbital wall originating in the lacrimal extends into the prefrontal but is significantly thinner and taller in this element. This wall is also present in Colobomycter pholeter (figure 4B in MacDougall et al., 2017b) as well as in a redescription of the bolosaurid Belebey vegrandis (figure 1 in Reisz et al., 2007). A shallow fossa is present on the anterior surface of the antorbital wall, composed of the lacrimal and prefrontal, which has been described in C. pholeter but not C. vaughni (MacDougall, Modesto & Reisz, 2016). Where the prefrontal and lacrimal suture to each other on this ridge, the prefrontal also forms a thin jagged contact with the palatine.

The frontal is a narrow and roughly rectangular element extending anteroposteriorly along the skull roof, with a medial suture between the two sides forming a jagged tongue-and-groove pattern (Figs. 2A, 2B and 4A) that is most pronounced in the midorbital region. While the external dorsal portion of the nasal-frontal suture extends obliquely from the midline of the skull, as described in Acleistorhinus pteroticus (deBraga & Reisz, 1996), the ventral internal portion is linear and extends transversely to the midline of the skull. In addition, the nasal-frontal sutures are somewhat asymmetrical, where the suture on the right side is more lateral in direction than the left, but the overall pattern remains the same, in contrast to what is described in Colobomycter pholeter (Modesto & Reisz, 2008). The shallow anterior ridge that runs along the ventral surface of the bone and likely contained the orbitonasal vein appears to end at the anterior orbit, although it remains longitudinal in direction. The prefrontal-frontal suture appears fairly simple on the external surface of the skull, and extends largely anteroposteriorly, curving laterally at its posterior end towards the orbit. A lateral supraorbital flange of the frontal separates the prefrontal and postfrontal, which is thin on both the dorsal and ventral surfaces. The posterior sutures of each side of the frontal are V-shaped on their dorsal surface, as with C. vaughni (MacDougall, Modesto & Reisz, 2016). The dorsal surface of the frontal is decorated by multiple prominent tuberosities and dimples. On the ventral surface of the skull, the frontal not only contributes to the anterior orbital ridge but also forms a concavity together with the prefrontal where the eye likely would have resided. Moreover, the frontal contribution to the anterior orbital ridge is contiguous with a pronounced ridge that extends longitudinally to the parietal. This ridge creates a deep concave channel on the ventral surface of the skull roof, likely separating the eye from the olfactory tract.

In contrast to the relative complexity of the frontal and prefrontal bones, the postfrontal is a relatively simple sheet-like bone on the dorsal surface of the skull, sutured anteriorly and medially to the frontal and posteriorly to the parietal and postorbital bones. In OMNH 73515, the postfrontal is partially preserved as a thin fragment on the left side of the skull and completely preserved on the right side (Figs. 2A, 2B and 4A). As described for Colobomycter vaughni (MacDougall, Modesto & Reisz, 2016), this posterodorsal element of the orbit is largely triradiate in outline with anterior, posterior and ventral processes. While the dorsal contact with the frontal is uneven and diagonal, the ventral portion of the anterior postfrontal extends almost to the prefrontal as a circular plate. As with the prefrontal and frontal, the postfrontal is dotted with tuberosities and dimples.

The postorbital bridges the dorsal and lateral surfaces of the skull roof and extends ventrally along the posterior orbital edge. Damage to the posterior region of the skull roof on the left side has resulted in an entirely disarticulated postorbital. The only identifiable fragment of the left postorbital is lying in the respective orbit. On the right side, the postorbital is complete and articulated with the postfrontal in an uneven suture pointing towards the pineal foramen (Figs. 2A, 2B and 4A). The suture between the postorbital and the dorsal edge of the jugal is smooth, with the lateroventral edge of the postorbital contributing to the temporal fenestra. This contribution to the dorsal edge of the temporal fenestra apparently distinguishes it from what is observed in Colobomycter pholeter and C. vaughni (MacDougall, Modesto & Reisz, 2016; MacDougall et al., 2017b), as well as in Acleistorhinus (deBraga & Reisz, 1996). Ornamentation of the dorsolateral surface of the postorbital is considerably more pronounced on the postorbital than in more anterior skull roof elements, with larger tuberosities and several deep dimples. The ventromedial surface is smooth, except for one small foramen immediately behind the posterior border of the orbit.

While previously illustrated in Reisz, Macdougall & Modesto (2014), the parietal has not been fully described. This posterodorsal skull roof element is broadly hexagonal in outline, and is bordered by the frontal anteriorly, the postfrontal anterolaterally and the postorbital laterally (Figs. 2A and 4A). The interparietal suture is tongue-and-groove in appearance, similar to the interfrontal suture but less jagged and uneven. A notable tongue-like projection of the left parietal overlies the right parietal and forms much of the anterior border of the pineal foramen. The large pineal foramen is located near the middle of the interparietal suture, similar to the condition seen in Colobomycter pholeter (figure 3A in MacDougall et al., 2017b). This condition contrasts to what is described in Acleistorhinus pteroticus, where the pineal foramen is positioned anteriorly in the interparietal suture (deBraga & Reisz, 1996). The jagged fronto-parietal suture extends along the posterior half of the orbit, with the parietal underlying the frontal significantly. Lateral parietal-postfrontal and parietal-postorbital sutures are wavelike, with a tongue-and-groove pattern similar to the interparietal suture. A groove on the posterolateral side of each side of the parietal would also underly the supratemporal. On the dorsal surface, tuberosities and dimples ornament the parietal in the same way as other dorsal skull elements. The posterior occipital surface of the parietal is relatively smooth, curving posterolaterally towards the parietal-supratemporal suture, and would most likely form attachment points for neck musculature. In contrast to the dorsal surface of the bone, the ventral surface is mostly smooth and slightly concave, with the exception of the ridged area adjacent to the ventral longitudinal ridge of the frontal, a posterior continuation of the massive ridge of the ventral surface of the latter bone. Several tiny foramina are present near the suture with the frontal. A shallow extension of the fronto-parietal ridge curves towards and terminates at the pineal foramen, but connect to very slender, lip-like flanges that form the ventrolateral edges of the foramen, and approach each other posteriorly. An additional pair of foramina occurs centrally on each side of the parietal, near the pineal foramen.

The postparietals are present and well-preserved along the skull midline in articulation with the parietals in OMNH 73515, suturing onto the parietal anteriorly and primarily contributing to the occipital surface of the skull (Figs. 2A and 4A). While the tiny skull roof contribution of the postparietal is bulbous, the occipital surface of the postparietal is concave and slightly rugose. In contrast, the ventral surface of the postparietal is relatively smooth. The right postparietal appears to be considerably larger than the one on the left, and the bulbous dorsal surface originates from this element but it appears that both elements underlie the parietal and are sutured to the ventral surface of that bone. Ventrally, the postparietals carry a pair of delicate ridges that extend from the posterior region of the parietal, near the pineal foramen, and are marked by the presence of a slender anterior process. These ridges broaden out as they extend into the main body of this element. The left tabular is complete and articulated with the posterolateral left parietal in OMNH 73515 but does not contact any other elements. It is very small and unornamented and would join with the parietal-supratemporal suture laterally in a fully articulated specimen.

The squamosal is a fairly large, heavily sculptured component of the skull roof that contacts the postorbital and the supratemporal medially, contributes to the border of the temporal opening anteriorly and overlies the quadratojugal ventrally. The right squamosal is nearly complete and articulated with the adjacent postorbital in OMNH 73515, but the left squamosal is very fragmented and has not been rendered (Figs. 2A, 2B and 4). This posterior skull roof element is trapezoidal with weakly convex and concave posterior and anterior margins, measuring approximately 1 cm in anteroposterior length. The ventral margin of the squamosal extends posteroventrally where it contacts the dorsal process of the quadratojugal. The anterodorsal margin of the squamosal forms a straight contact with the postorbital, curving around a large tuberosity in the middle of the squamosal. At the dorsal border of this element, a small shelf extends dorsally that underlies the postorbital and would underly the supratemporal in a fully articulated specimen. This ridge continues onto the posterior surface, where it wraps around to the occipital region as in Colobomycter pholeter (figure 3D in MacDougall et al., 2017b). Posteroventrally, the ridge also contributes to the dorsal border of the quadrate foramen.

As in most parareptiles the quadratojugal is tall and occupies a significant portion of the postorbital region of the skull roof. As with the squamosal, the right quadratojugal of OMNH 73515 is complete and partially articulated while the left is heavily damaged (Figs. 2A, 2B and 4B). If the right quadratojugal were fully articulated, it would suture onto the squamosal dorsally, the jugal anteriorly and the quadrate medially. The anterior process of the quadratojugal is triangular in shape, contributing to the ventral surface of the skull as well as a slightly jagged posteroventral border of the temporal fenestra. While the quadratojugal possesses tuberculous ornamentation on the lateral surface, it is much less pronounced than its neighbouring elements. In contrast, the external surface of the occipital flange that contributes to the quadrate foramen, as well as the process supporting the lateral condyle of the quadrate, is smooth. A small foramen is present on the dorsomedial surface of the quadratojugal, but otherwise the internal surface is smooth.

The supratemporal is also a major component of the posterolateral region of the skull roof, but due to postmortem fragmentation the left supratemporal is missing from OMNH 73515 (Fig. 4A). The right supratemporal is disarticulated and only partially preserved, but still associated with the specimen at the lateroventral right corner of the occipital surface. On its dorsal surface, the supratemporal is adorned with short, wide tuberosities, whereas the ventral surface is smooth and slightly concave. This element forms much of the posteroventral skull roof and is underlain by the parietal medially, the postorbital anteriorly and the squamosal laterally with particularly large facets of articulation on the parietal bone. While the supratemporal appears to readily fall off postmortem in many acleistorhinids, it clearly demonstrates that this bone made a significant contribution to the ventral skull roof surface. This extensive ventral skull roof contribution contribution is observed in Macroleter and Saurodectes (Reisz & Scott, 2002; Tsuji, 2006), indicating that this is a widespread condition in parareptiles, in contrast to its restricted contribution or absence in eureptiles and most synapsids.

Palate

In Delorhynchus cifellii, the palate is comprised of four dentigerous elements—the vomer, the palatine, the pterygoid and the ectopterygoid (Fig. 5). Much of the ventral palatal surface is covered by fields of teeth arranged in distinct groups that extend over several bones and separated from each tooth by slightly concave areas and distinct large grooves. Anteriorly, the choana is long and slender, separating the vomer from the maxilla and ending posteriorly at the curved lateral process of the palatine bone. Remarkably, the choana is connected posteriorly to a wide groove that extends posterolaterally along the ventral surface of the palatine from the choana parallel to the tooth row and the lateral edge of the palatine. This groove changes direction at the palatine-ectopterygoid suture and continues in a posteromedial direction across the ventral surfaces of the ectopterygoid and pterygoid. Posteriorly, this groove separates the dental field of the transverse flange and its posterior continuation on the quadrate ramus of the pterygoid from the basicranial articulation. It is possible that this groove extending from the choana into the throat region may represent an air passage that allowed for respiration even when the palatal dentition was engaged in holding prey.

Figure 5 Palate and braincase of Delorhynchus cifellii, OMNH 73515.

(A) Dorsal, (B) lateral and (C) ventral views. Abbreviations: bh, basihyal; bo, basioccipital; ch, ceratohyal; ect, ectopterygoid; epi, epipterygoid; ex, exoccipital; nc, neurocranium; pal, palatine; pbs, parabasisphenoid; pt, pterygoid; q, quadrate; s, stapes; sph, sphenethmoid; v, vomer.

The vomer is triangular in outline, forming the anterior tip of the palatal region (Fig. 5). This palatal element is bordered posterolaterally by the palatine and posteromedially by the narrow anterior process of the pterygoid, with the palatine-vomer suture at the level of the interpterygoid vacuity as seen in Acleistorhinus (deBraga & Reisz, 1996). Two fields of vomerine teeth are present on the ventral surface, with a narrower lateral field and a wider medial field as seen in other acleistorhinids as well as in Macroleter and Saurodektes (Reisz & Scott, 2002; Tsuji, 2006). Where the tooth rows converge at the anterior terminus, the teeth are comparatively larger in size as described in Colobomycter pholeter and Macroleter (MacDougall et al., 2017b). Much of the dorsal surface is smooth, except for an alar flange that connects to the orbitonasal ridge of the palatine. In contrast to Macroleter and Captorhinus laticeps (Heaton, 1979; Tsuji, 2006), the alar flange is entirely straight, forming the medial border of the internal nares. A well developed, tall medial flange is an anterior extension of a similar flange along the medial edge of the pterygoid, and extends anteriorly to the premaxilla. This flange forms a relatively thick suture between the left and right vomers.

As with Colobomycter pholeter (figure 3B in MacDougall et al., 2017b), much of the lateral part of the palate consists of the palatine (Fig. 5). This element is bordered by the vomer anteromedially, forming the posterior choana, and contacts the pterygoid medially and the ectopterygoid posterolaterally. The lateral contact between the ectopterygoid and palatine is interrupted by the suborbital foramen, which is modest on the left side and widened on the right side by a small hole. A large dental field is present diagonally along much of the length of the palatine and extends onto the pterygoid as with other acleistorhinids (deBraga & Reisz, 1996). This field widens considerably along its length, decreasing in the size of the teeth posteriorly but increasing in density. A wide groove with a smooth surface extending from the posterior choana is bordered medially by this tooth field, and laterally by a thickened ridge contacting the maxilla. On its dorsal surface, the alar flange continues into the transverse orbitonasal ridge as seen in Captorhinus laticeps and Macroleter poezicus (Heaton, 1979; Tsuji, 2006). These two ridges are tallest at their contact, suturing onto the lacrimal and prefrontal. A shallow, wide ridge extends posterolaterally, sutured to the maxilla.

The ectopterygoid is a relatively large element of the palate, bridging a large gap between the posterior region of the palatine and the transverse flange of the pterygoid. Both ectopterygoids are present in OMNH 73515 and complete, but the right ectopterygoid is slightly disarticulated (Figs. 5A and 5C). This element is broadly triangular, suturing onto the palatine anteromedially, the pterygoid posteromedially and the maxilla laterally. The wide groove extending from the ventral surface of the palatine curves sharply on the ectopterygoid, continuing posteromedially into the pterygoid. This groove, likely serving as an air passage, is bordered by several palatal teeth medially and by thickened contacts with the maxilla and jugal laterally. Palatal teeth are present on the ectopterygoid as part of the relatively wide field in this region of the palate, which differs from the condition described in both Colobomycter pholeter and Acleistorhinus pteroticus (deBraga & Reisz, 1996; MacDougall et al., 2017b). The dorsal surface is mostly smooth, with a shallow ridge resulting from the canal.

The pterygoid is the largest element of the palate as with most early Permian amniotes, including acleistorhinids, with a broadly triangular shape (Fig. 5). This element possesses a mostly broad, flattened palatal process, a concave transverse process and a quadrate ramus extending posterolaterally. The anterior palatal process extends anteriorly to the vomer and palatine, separating the posterior half of the vomers. Two fields of teeth are present on this anterior process, with one anterolateral field extending from its central portion onto the palatine and an anteromedial field. The anteromedial field of teeth can be subdivided into a row that extends along the medial edge of the bone where the interpterygoid vacuity is located, and a secondary row that extends roughly parallel to the medial row. This pattern resembles what is seen in Colobomycter (MacDougall et al., 2017b). The transverse flange is a robust feature of the bone, and it extends an edentulous process anterolaterally towards the medial process of the jugal, with a thin bifurcating sutural contact with the latter. The rest of the ventral surface of the transverse flange is highly dentigerous, with a row of large teeth on the posteroventral edge of the transverse process as well numerous smaller teeth as in C. pholeter (figure 3B in MacDougall et al., 2017b). Anteromedially, the transverse flange borders the broad groove that separates the dentigerous portion of the flange from the palatal region of the pterygoid, and both the transverse flange and the groove extend posteromedially into the quadrate ramus. In contrast to the anterior and transverse processes, the quadrate ramus is relatively smooth and curves posterolaterally to cup the quadrate. As in other acleistorhinids, the row of marginal teeth present on the posterior edge of the transverse flange continues into the quadrate ramus, decreasing in size as they go along. The dorsal surface of the pterygoid is less complicated, with two noteworthy features: a paired medial ridge extending dorsally from the tip of the anterior process to the orbital region where the sphenethmoid is located, and a groove and sutural area in the region of the basicranial articulation for attachment of the epipterygoid ossifications, which are preserved in articulation. Posterodorsally, a tympanic ramus is present on the quadrate flange of the left pterygoid.

The morphology of the epipterygoid, while comparatively more delicate, resembles the condition seen in the redescription of Macroleter poezicus (Tsuji, 2006). Anteroventrally, the base of the epipterygoid contributes with the pterygoid to the articular surface of the basicranial articulation, while posteriorly it overlies the anteromedial quadrate ramus of the pterygoid (Fig. 5B). The dorsal process of the epipterygoid, preserved in its entirety, extends as a slender pillar posteromedially. As with the quadrate ramus of the pterygoid, the surface of the epipterygoid is smooth.

In Delorhynchus cifellii, the parasphenoid and basisphenoid are fused together and henceforth are referred to as the parabasisphenoid, with specific features that can be identified as part of the parasphenoid and basisphenoid will be described separately. The parabasisphenoid is unusually large and occupies the posterior half of the palatal surface, whereas in Acleistorhinus pteroticus it appears relatively shorter and occupies only one third of the palatal length (figure 1B in deBraga & Reisz, 1996). As with A. pteroticus and Colobomycter pholeter (figure 3B in MacDougall et al., 2017b), this posteroventral palatal element is triangular in outline (Figs. 5C and 6). A narrow, short cultriform process extends anteriorly, and overlies the interpterygoid vacuity. Given the length of the cultriform process, the anteriormost extent of the interpterygoid vacuity was not bisected by the cultriform process. Where the cultriform process merges with the rest of the parabasisphenoid, two fields of short teeth are present on the ventral surface. These tooth rows diverge posteriorly resembling the condition seen in the redescription of A. pteroticus and Lanthanosuchus watsoni (deBraga & Reisz, 1996). The most lateral rows of teeth are positioned over the cristae ventrolaterales along the main body of the parasphenoid, whereas the most medial rows are present on low ridges as in L. watsoni (deBraga & Reisz, 1996). These rows of teeth extend to the level of the posterior termination of the pterygoid teeth on the quadrate ramus. Unlike A. pteroticus, the tooth rows also extend anterolaterally to overly the basisphenoid rostrum and posterior cultriform process. The basipterygoid processes extend anterolaterally and would form synovial joints with the basicranial recesses. As noted in the redescription of Macroleter poezicus (Tsuji, 2006), the sella turcica is positioned caudally to the dorsal body, bordered by the clinoid processes posterolaterally. The medial and lateral surfaces of each clinoid process is smooth, and the foramen for cranial nerve VI pierces each process posterodorsally. Transversely, the clinoid processes are connected by a thin dorsum sellae. A pair of ridges overly the cristae ventrolateralis and extend posterolaterally.

Figure 6 Parabasisphenoid and sphenethmoid of Delorhynchus cifellii, OMNH 73515.

(A) Dorsal, (B) lateral and (C) ventral views. Abbreviations: bpt. tub., basipterygoid tuberosity; cult. pr., cultriform process; d. pr., dorsal process of sphenethmoid; d. s., dorsum sellae; pbs, parabasisphenoid; pr. sell., processus sellaris; sph, sphenethmoid; v. k., ventral keel; vid. sul., vidian sulcus.

Both quadrates are preserved in the holotype, with the right element in articulation with the pterygoid. As in Acleistorhinus (deBraga & Reisz, 1996), the quadrate is short anteroposteriorly, overlying the lateral quadrate ramus of the pterygoid as a thin triangle of bone (Fig. 4), and with a massive transversely oriented condyle. The quadrate is slender anteriorly but thickens considerably as it extends posteriorly, contributing to the ventral border of the quadrate foramen and suturing onto the quadratojugal laterally. Posterolaterally, the quadrate underlies the quadratojugal as a massive process. As in Colobomycter pholeter (MacDougall et al., 2017b), the contact with the articular facet would be secure in a fully articulated individual, with protuberances of the quadrate condyle fitting into the dorsal sockets of the articular. Medially, a short, curved process of the quadrate extends anteromedially, providing an articulating or supporting surface for the stapes. The dorsal process of the quadrate is tall and is exposed occipitally, with a bladelike lateral extension and an open dorsal termination. This process curves medially and separates the squamosal and pterygoid.

The sphenethmoid is preserved in this specimen as an elongated Y-shaped sheet of bone, as described in Captorhinus laticeps as the interorbital septum (Heaton, 1979). This element is shifted out of place, with the ventral margin partially contacting the cultriform process of the parabasisphenoid (Figs. 5A, 5B and 6). In a fully articulated specimen, the ventral margin of the sphenethmoid would likely contact the trough of the cultriform process. Unlike Ca. laticeps, the bifurcation into the sola supraseptales occurs at approximately one-third of the height of the sphenethmoid. The surface of this element is smooth as was described in Co. pholeter (Modesto & Reisz, 2008). The sola supraseptales are somewhat damaged, but the preserved margin is thickened dorsally and would likely contact the mediorbital ridge of the prefrontal, frontal and postfrontal as suggested for Ca. laticeps. The morphology of the sphenethmoid in Delorhynchus cifellii differs considerably from those seen in Dimetrodon and Ophiacodon (Romer & Price, 1940), where the sola supraseptales bifurcate at two-thirds of the height of the element. In addition, the anterolateral surface in these synapsids is noticeably sculptured and a pair of foramina extend anterolaterally through the base of the sola supraseptales, features not shared by D. cifellii.

Computer tomography of the skull has full exposed the stapes, which is completely preserved but disarticulated on both sides of OMNH 73515 (Figs. 5C and 7). This element is compact and robust, in contrast to the more elongated condition seen in captorhinids such as Captorhinus laticeps and Euconcordia cunninghami (Heaton, 1979; Reisz, Haridy & Müller, 2016) and synapsids such as Ophiacodon retroversus, Dimetrodon limbatus, Mesenosaurus romeri and Martensius bromackerensis (Romer & Price, 1940; Reisz & Berman, 2001). A large footplate is present as described in Acleistorhinus pteroticus (deBraga & Reisz, 1996), with a deep central pit, a thin dorsal edge and a broad ventral edge. The thickness of the footplate rim contrasts to what is observed in C. laticeps, where the rim is uniformly thin. Anteriorly, the footplate would contact the basisphenoid and prootic, whereas the posteromedial edge would lie within the stapedial recess of the basioccipital and opisthotic as in C. laticeps. The columella is short, broad and distally expanded in dorsal view, with ridges on its anterior, dorsal and ventral surfaces as with C. laticeps. Lateral expansion is observable from an occipital view, as in the redescription of Milleretta rubidgei (Gow, 1972). The robust morphology of the columella differs considerably from the elongated condition seen in captorhinids and synapsids (Romer & Price, 1940; Heaton, 1979; Reisz & Berman, 2001; Reisz, Haridy & Müller, 2016; Berman et al., 2020). Extending from the lateral end of the terminus to the dorsal process is the dorsal process, increasing in height medially and ending in unfinished bone that would likely connect to cartilage, as in the redescription of Milleretta rubidgei (Gow, 1972). The distal columella shares this unfinished texture, connecting to the lateral end of the dorsal process anteriorly, and would contact the quadrate in an interlocking process. Compared to both captorhinids and synapsids, the dorsal process is short and stout, without a pronounced channel for the vena capitis lateralis or the hyoid ramus of the facial nerve. The anterior ridge extends posteromedially from the ventrolateral end of the columella to the base of the footplate, but is not perforated by a dorsoventral foramen as in C. laticeps. The stapedial foramen instead extends anteroposteriorly immediately adjacent to the footplate, as observed in the redescription of Mesenosaurus romeri (Reisz & Berman, 2001). While this foramen is primarily anteroposterior in O. retroversus and D. limbatus, the orientation is comparatively more posteroventral in orientation (Romer & Price, 1940). It is likely that the stapedial artery was mostly or entirely directed anteroposteriorly.

Figure 7 Right stapes of Delorhynchus cifellii, OMNH 73515.

(A) Anterior, (B) dorsal and (C) occipital views. Abbreviations: d. pr., dorsal process of stapes; fp, footplate; s. for., stapedial foramen; s. sh., stapedial shaft.

Immediately posterior to the parabasisphenoid, the basioccipital-exoccipital complex forms the occipital condyle in a fully articulated specimen (Fig. 5). The basioccipital is roughly pentagonal in shape, with a thickened posterior surface where it sutures onto the paired exoccipitals. The exoccipitals extend dorsally to form the dorsolateral borders of the occipital condyle and the ventrolateral edges of the foramen magnum. While the columnar shape of these elements generally resembles the condition of Macroleter poezicus (Tsuji, 2006), the dorsal surface lacks medial and lateral processes. The shape of the paired exoccipitals therefore more closely resembles what is reconstructed for Saurodektes kitchingorum by Reisz & Scott (2002).

Partially disarticulated within the posterior region of the skull is a neurocranial complex consisting of the supraoccipital, opisthotic and prootic complex (Figs. 5 and 8). These three elements are fused in Delorhynchus cifellii without visible sutures, and the following descriptions of the neurocranium elements therefore reference their probable anatomy. However, features such as the paired posttemporal foramina dividing the supraoccipital and opisthotic and the venous notch allow for some differentiation of individual elements. As observed in the redescription of Macroleter poezicus (Tsuji, 2006), the supraoccipital is anteroventrally concave, with a low median sagittal ridge and a semicircular crest on the occipital surface. Unlike Macroleter, the semicircular crest is bifurcated by the sagittal ridge, which would connect with the ventral corner of the paired postparietals. Occipitally, the supraoccipital is concave and forms the dorsal border of the foramen magnum, with a medial notch resulting in a keyhole shape similar to the condition in Macroleter (Tsuji, 2006). Posteriorly, this braincase element includes a pair of wide, flattened indentations that would contact the dorsal exoccipitals. The prootic extends forward as paired, lateromedially curved processes which would contact the basisphenoid as in Milleretta (Gow, 1972) and Macroleter (Tsuji, 2006). The anterodorsal surfaces of these processes are marked by venous notches, as described in Dimetrodon limbatus by Romer & Price (1940) but more pronounced in concavity. Immediately ventral to the venous notches, a pair of foramina extend posteromedially through the middle of the prootic which would have probably enveloped cranial nerve VII as noted in Milleretta and Macroleter. Dorsally, the prootic is marked by paired cylindrical swellings which envelop the anterior semicircular canals. Interestingly, the largest element in the neurocranial complex is the opisthotic, which expands posterolaterally into dorsal and ventral regions of the paroccipital process. This morphology resembles what is seen in other parareptiles like Macroleter and Acleistorhinus, where the paroccipital process extends ventrolaterally to the skull roof. On the dorsal region of the paroccipital process, a shallow groove is present which may have connected to the supratemporal and parietal as noted in deBraga & Reisz (1996).

Figure 8 Supraoccipital-prootic-opisthotic complex of Delorhynchus cifellii, OMNH 73515.

(A) Anterior, (B) dorsal and (C) occipital views. Abbreviations: op, opisthotic; par, paroccipital processes of opisthotic; pro, prootic; sup, supraoccipital; ven, venous notches.

Ventrally, the neurocranium is relatively open, with partially ossified channels for the bony labyrinth (Fig. 9). The vestibule was only reconstructed at its contact with the posterior semicircular canal (PSC) due to the disarticulation of the neurocranium and the lack of ossification. Neurocranial morphology indicates that the entire bony labyrinth was tilted posteroventrally. Anteriorly, the prootic encapsulates the majority of the anterior semicircular canals (ASC) and the anterior and lateral ampullae, while posterolaterally the opisthotic envelops the lateral semicircular canals (LSC), the secondary common crus and the lateral region of the posterior semicircular canal. It is likely that the opisthotic would also have encapsulated the vestibule dorsally. The only feature of the bony labyrinth present within the supraoccipital is the common crus and the medial sections of the ASC and PSC. Given the orientation of the neurocranial complex, the ASC would be positioned dorsally even without the curvature of the common crus. Both the ASC and PSC are arcuate, while the LSC is L-shaped. Anterodorsally, the dorsal expansion of the anterior semicircular canal is slight as with varanopids (Bazzana et al., 2021). A secondary common crus is not present, as the LSC extends posteromedially towards the posterior semicircular canal but does not connect with it. Both the angle between the ASC and PSC and the angle between the ASC and LSC are sub-orthogonal, with averaged measurements of 71° and 87° respectively. While the angle between the ASC and the LSC is the same as in Mesenosaurus, the angle between the ASC and PSC is less orthogonal than in varanopids (Bazzana et al., 2021).

Figure 9 Semicircular canals of Delorhynchus cifellii, OMNH 73515.

(A) Anterior view, (B) lateral view and (C) occipital view of neurocranium, (D) anterior view, (E) lateral view, (F) dorsal view and (G) occipital view of left semicircular canals. Abbreviations: aasc, ampulla of anterior semicircular canal; alsc, ampulla of lateral semicircular canal; asc, anterior semicircular canal; cc, common crus; lsc, lateral semicircular canal; op, opisthotic; par, paroccipital process of opisthotic; pro, prootic; psc, posterior semicircular canal; sup, supraoccipital; vs, vestibule.

The basihyal and ceratohyals are preserved in OMNH 73515, but are disarticulated from one another (Fig. 5B). The basihyal possesses a semilunate anterior edge and narrows into paired posterior projections, as opposed to the hourglass shape observed in Saurodektes kitchingorum (Reisz & Scott, 2002). While the ventral surface is smooth, a pair of rounded protrusions are present on the dorsal surface near the posterior processes. The ceratohyals are elongated and rodlike that expand distally, with slight medial curvature as in S. kitchingorum.

Mandible

The anatomy of the lower jaw corresponds to what has been previously described for Delorhynchus cifellii in Haridy, Macdougall & Reisz (2018). The dentary in OMNH 73515 is mostly complete and articulated with adjacent mandibular elements, but a break at the tip of the mandible disarticulates the left side of the mandible from the rest of the skull (Figs. 2C and 10). Anteriorly, the dentary is underlain by the splenial in a relatively smooth suture while the posterolingual dentary sutures onto the angular in a similar pattern. The posterolateral surface of the dentary is adjacent to, and overlies, the anterior surangular. Both coronoid elements are also supported by the dorsomedial surface of the dentary. On the anterior surface of the dentary, the lateral surface is dotted with small foramina, a pattern shared by Colobomycter (MacDougall, Modesto & Reisz, 2016), Karutia fortunata (Cisneros et al., 2021) and D. multidentatus (Rowe et al., 2021). A posterodorsal extension of the dentary extends over the lateral surface of the surangular as previously described in Reisz, Macdougall & Modesto (2014), a feature not found in other acleistorhinids. The ventrolateral surface of the dentary also bifurcates posteriorly at the contact with the angular. On its medial surface, the dentary forms the lateral border of the Meckelian canal, which is deeply concave resulting in a transverse C-shape. The Meckelian canal widens posteriorly and extends into the posterior mandibular elements. Immediately anterior to the coronoid, a deep foramen branches off from the Meckelian canal. This foramen extends anteriorly directly underneath the dentigerous section of the dentary to the mandibular symphysis, and may have provided blood supply to growing teeth.

Figure 10 Mandible of Delorhynchus cifellii, OMNH 73515.

(A) Dorsal view of mandible, (B) medial view of right mandible and (C) lateral view of right mandible. Abbreviations: an, angular; art, articular; cr1, first coronoid ossification; cr2, second coronoid ossification; d, dentary; p. Mck. f., posterior Meckelian fossa; pra, prearticular; sur, surangular.

In this specimen of Delorhynchus cifellii, the coronoid complex is complete and articulated on each side of the mandible, with the suture between the anterior and posterior coronoid ossifications clearly visible as first described in Haridy, Macdougall & Reisz (2018). These ossifications suture onto the junction between the splenial and dentary (Figs. 10A and 10B). The dorsal surfaces of the coronoids are covered with shagreens of small, short teeth. As with other specimens of Delorhynchus cifellii, as well as D. multidentatus, the posterior coronoid ossifications terminate in winglike projections that extend dorsomedially, adjacent to the coronoid process.

The splenial underlies the dentary immediately behind the dental symphysis and extends posteriorly to the coronoid process, sheathing the Meckelian canal lingually (Fig. 10). The dorsal splenial-dentary suture directly underlies the coronoid ossifications, diverging at the midpoint of the posterior coronoid and extending posteroventrally towards the lingual surface of the angular. Anterodorsally, the splenial bifurcates into a short dorsal projection and a longer ventral projection that extends to the dental symphysis. The ventral splenial contacts the dentary anteriorly and forms the posteriormost extent of this mandibular element.

The angular is a curved sheet extending from the anteriormost coronoid to the articulating surface of the mandible (Figs. 10A and 10B). Examination of the inside of the anterior mandible reveals a thin process of the angular sheathing the Meckelian canal ventrally, which increases in concavity. This process is also covered by the dentary laterally near the intercoronoid suture, where it appears on the lateral surface of the mandible as a triangular process. The posterior angular also extends to the articulating facet of the surangular. Immediately dorsal to the posterior terminus of the splenial, an oval-shaped foramen perforates the lingual surface of the angular. This foramen is bordered by the angular ventrally and the prearticular dorsally, and a small extension of the splenial overlies the anterior edge. The posterior extension of the angular forms the ventral portion of the mandible, curving lingually with a well-sculpted lateral surface. The cup-shaped dorsolingual surface of the angular retains its shape to its posterior terminus, where it sutures the articulating facet of the mandible.

As first described in Reisz, Macdougall & Modesto (2014), much of the coronoid process is made up of the surangular (Fig. 10). This dorsal mandibular element underlies the dentary anteriorly as a thin sheet before expanding into the posteriormost labial surface of the mandible. The thin posterior terminus of the dentary overlying the surangular continues into a lateral ridge of the surangular. This ridge extends to the posterior bifurcation of this element, where a lingual extension cups the articular anteriorly. Other than several small foramina near the bifurcation of the surangular, the lingual surface is smooth and laterally convex.

The dorsolingual surface of the posterior part of the mandible is covered by the prearticular, which is more heavily sculpted on its lingual surface than most other mandibular elements (Figs. 10A and 10B). Two ridges are present, with an anterior ridge and a well-defined dorsal ridge extending posteroventrally. As with the surangular, the prearticular cups the articular on its posterior terminus. The ventral surface of this element also contacts the angular in a mostly straight suture that curves ventrally under the articular suture. Unlike OMNH 72363 (Haridy, Macdougall & Reisz, 2018), the prearticular does not contact the anterior coronoid, and these differences may be due to ontogeny.

The articular is the most heavily built element of the mandible, with a wrinkled external surface (Figs. 10A and 10B). Compared to the thoroughly described articular in Captorhinus laticeps, this element is much shorter (Fox & Bowman, 1966). An anterior extension of this element with thin walls forms the posterior wall of the Meckelian canal, although this acuminate extension is shorter than what was seen in Captorhinus laticeps. As with Acleistorhinus pteroticus and Colobomycter pholeter (MacDougall et al., 2017b), there are two articulating facets on the posterodorsal surface. Two pairs of foramina can be observed on the lingual articulating facet—a small pair on the posterior surface matching the condition in A. pteroticus (Daly, 1969), and a larger pair on the lingual surface.

Postcranium

The holotype of Delorhynchus cifellii includes numerous postcranial elements, permitting a better understanding of the basic acleistorhinid anatomy. A nearly complete atlas-axis complex is present in this parareptile, including a proatlas, as well as other cervical and dorsal vertebrae. A proatlas is preserved in the form of the left proatlantal neural arch, which has been dislodged from its contact with the atlantal neural arch. The oval-shaped plate and thin, posteriorly projecting spine more closely resembles the condition of Captorhinus aguti as described in Fox & Bowman (1966) than the triangular process present in Ophiacodon and Dimetrodon (Romer & Price, 1940). Anteriorly, the proatlantal neural arch widens at its shallowly concave contact with the exoccipital. While the general morphology of the proatlantal neural spine resembles C. aguti, the spine appears to be slightly more elongated. The postzygapophosis is short and robust, with a rounded, shallowly concave surface similar to the exoccipital-proatlas contact where it would suture onto the atlas. The atlas is preserved but the atlantal neural arch has been disarticulated from the atlantal centrum (Fig. 1B). The atlantal centrum is robust, with short laterally projecting processes which may have partially supported the axial rib as mentioned in Dilkes & Reisz (1986) despite the absence of the axial intercentrum. Dorsally, the atlantal centrum is flattened where it contributes to the vertebral foramen. Posteriorly, the notochordal canal of the atlantal centrum expands posteriorly. Anteroventrally, the foot processes are robust with flattened atlantal prezygapophyses that would support the proatlantal postzygapophyses. Wide, laterally projecting atlantal postzygaphophyses are present on the neural plates of the atlantal neural arch. A pair of bifurcated neural spines project from the postzygapophoses, which are perforated on each side by a small dorsally oriented foramen. An atlantal intercentrum is also present, but has been disarticulated and partially obscures the atlantal centrum.

Preserved portions of the axis include the axial centrum and the axial neural arch (Fig. 11A). The anterior end of the axial centrum is shallowly slanted posteriorly, which would have accommodated the axial intercentrum. Ventrally, a small divot is present which which would have contacted the axial intercentrum, indicating that the axial intercentrum would have been positioned underneath the atlantal pleurocentrum as in captorhinids such as Captorhinus aguti and C. laticeps (Fox & Bowman, 1966; Dilkes & Reisz, 1986). This divot extends as a small ridge to the posterior end of the axial centrum, which results in a pinched ventral surface. The parapophyses are present on the axial centrum but do not extend onto the axial neural arch to the extend observed in C. aguti (figure 36 in Fox & Bowman, 1966). Wide, prominent zygapophyses are present on the neural arch, with morphology that closely resembles C. aguti. The axial neural spine is tall and possesses an anteroventral projection, as with eureptiles and synapsids. In contrast to the concave posterior base of the axial neural spine in synapsids (Romer & Price, 1940), the neural spine of Delorhynchus cifellii is smooth.

Figure 11 Representative vertebrae of Delorhynchus cifellii, OMNH 73515 in anterior, lateral, dorsal, and ventral views.

(A) Axis vertebra, (B) third cervical vertebra, (C) sixth cervical vertebra and (D) anterior dorsal vertebra.

Excluding the atlas-axis complex, a total of four other cervical vertebrae are present in this specimen and are mostly articulated with one another (Figs. 11B and 11C). This assessment is based partly upon the cervical ribs associated with the specimen, as well as the morphology of the neural spines. While the neural spines of the third and fourth cervicals are posterodorsally directed and approximately one-half the length of the vertebrae, the spines on the fifth and sixth vertebrae are dramatically different in shape. The fifth neural spine is considerably shortened and posteriorly directed, while the sixth neural spine is almost nonexistent and projects anterodorsally. The cervical centra are also strongly pinched in comparison to the dorsals, with the sixth centrum displaying a more intermediate condition. In Captorhinus cervicals, based upon personal observation, the third and fourth cervicals display similar pinched texture on their ventral surfaces. However, the fifth and sixth vertebrae, as well as the dorsal vertebrae, do not display this pattern. The prezygapophyses are flattened, dorsally concave and mediolaterally broad. Compared to the prezygapophyses, the postzygapophyses are mediolaterally thinner, dorsally convex and project posteriorly. The zygapophyses as a whole are mediolaterally tilted towards the midline, as is described in C. aguti by Fox & Bowman (1966). Transverse processes are oriented lateroventrally, tapering as they extend anteroventrally. On the fifth and sixth cervicals, a small, rounded pathology protrudes between the zygapophyses on the right side of each vertebra (Fig. 11C), with the pathology on the fifth cervical vertebra being larger than on the sixth.

Nineteen dorsal vertebrae are preserved in OMNH 73515, which would result in a minimum count of twenty-five presacral vertebrae (Fig. 11D) in contrast to the nineteen exposed and described in Reisz, Macdougall & Modesto (2014). It is likely that these vertebrae represent most of the presacral series. In comparison to the cervical vertebrae, the dorsal centra are less pinched. Dorsal neural spines are uniformly one-third the length of the corresponding vertebrae where preserved. The dorsal zygapophyses closely resemble the cervical zygapophyses, with the exceptions of shallower mediolateral tilting and flattened dorsal surfaces. At the posterior end of the prezygapophyses, the dorsal neural arches are sharply pinched inwards. The transverse processes are laterally oriented and proportionately shorter than in the cervical vertebrae.

Four cervical ribs are associated with this skeleton, with three from the left side and one on the right (Fig. 1B). The partial articulation of these ribs with cervical vertebrae has led to tentative identification of the fourth left, fifth left and right and sixth left cervical ribs. While the fourth left cervical rib is shortened, triangular and tapers ventrally, the fifth and sixth cervical ribs are relatively elongated and expand ventrally. The dorsal ribs are jumbled together and preserved in varying degrees of completeness, with a total of twenty-five dorsal ends associated with this specimen. Unlike the posterior cervical ribs, the dorsal ribs are ventrally slender and are open at their ventral ends.

As with other early Permian reptiles and some synapsids, the paired clavicle consists of a broadened ventral plate overlying the anterior coracoid and interclavicle and a slender dorsal projection that would cap the anterior scapula (Figs. 12A and 12B). As with Captorhinus aguti (Fox & Bowman, 1966; Holmes, 1977), a narrow flange would have slightly enveloped the lateral surface of the scapula, which continues into the ventral plate. Towards the tip of the dorsal projection, a shallow groove runs through the scapular surface which may have marked the suprascapular. The flange extends ventrally and merges into the broad ventral plate, which is notched posteroventrally where it would contact the interclavicle. This flange is thinner than what is observed in C. aguti by Fox & Bowman (1966) and Holmes (1977) and lacks a posterior process.

Figure 12 Shoulder girdle elements of Delorhynchus cifellii, OMNH 73515.

(A) Lateral view and (B) medial view of left clavicle, (C) dorsal and (D) ventral views of interclavicle, (E) lateral view and (F) medial view of left scapulocoracoid. Abbreviations: gl, glenoid; gl. for., glenoid foramen; sg. for., supraglenoid foramen.

The interclavicle is a T-shaped structure, with a crossbar and stem that most closely resembles the condition in Milleretta (Gow, 1972). The ventral portion of the crossbar possesses a thin border projecting anteriorly which would have underlain the clavicle, although the posterolateral section of the border is not inset into the thickened crossbar as in Captorhinus aguti (Fox & Bowman, 1966). The medial border is raised slightly, which would have divided the paired clavicles (Figs. 12C and 12D), as in Milleretta (Gow, 1972). Both the dorsal and ventral surfaces of the interclavicle are otherwise smooth. The crossbar and stem of the crossbar are positioned perpendicular to one another as previously described in Milleretta (Gow, 1972). This condition contrasts sharply to Mesosaurus tenuidens, where the crossbar of the interclavicle gently merges with the stem (Modesto, 2010).

The scapulocoracoids are superficially similar to those in eureptiles like Captorhinus aguti, but with L-shaped plates resembling those of Mesosaurus tenuidens (Fox & Bowman, 1966; Holmes, 1977; Modesto, 2010). Segmentation of these elements has revealed that the left scapulocoracoid overlaps the right. General morphology can still be described for the scapula and coracoid, but the borders between elements are nearly indistinguishable, with the exception of a shallow suture on the anterior projection of the glenoid fossa (Figs. 12E and 12F). Whereas the anterior scapula is damaged on both sides, the posterior portion of the scapula is curved anteriorly and comparatively broader than what is described in C. aguti. The posterior border of the scapula extends forward slightly resulting in a curved dorsal edge. This condition resembles what is seen in M. tenuidens (Modesto, 2010), albeit markedly less pronounced. The glenoid buttress resembles the condition of other terrestrial tetrapods but the supraglenoid foramen does not perforate the glenoid buttress. Ventromedially, the coracoid possesses a broad plate extending throughout the entire length of the scapulocoracoid element. The glenoid fossa is the most robust section of the scapulocoracoid, with convex scapular and coracoid projections. Of the two lateral projections in the glenoid fossa, the anterior process is twice as tall as the posterior process and is pierced ventrally by the coracoid foramen. A prominent triceps process arises from the posterior coracoid, extending posterodorsally with a similar length to the posterior projection of the glenoid fossa. Medially, the scapulocoracoid is supported by several ridges as in C. aguti, although these ridges expand distally. The coracoid foramen extends anteriorly between the dorsal and anteroventral ridges.

The tetraradiate morphology of the humerus found in Permo-Carboniferous tetrapods is shared by Delorhynchus cifellii (Fig. 13). The proximal and distal ends of the humerus are twisted at approximate right angles to one another, although the angle is slightly closer to 90° than in Captorhinus aguti as noted in Fox & Bowman (1966). The synapsid condition is remarkably different, where the twist between proximal and distal ends is between 35° and 60° (Romer & Price, 1940). On the proximal end of this limb element, the deltopectoral crest projects anteromedially with a shallow dorsal depression. In C. aguti, this feature is significantly shorter, and is angled medioventrally with a flattened dorsal surface as described in Fox & Bowman (1966). Synapsids such as Ophiacodon also possess medioventrally oriented deltopectoral crests, but the angle between the crest and the proximal head is considerably sharper (Romer & Price, 1940). The proximal surface of articulation of the humerus is porous as in C. aguti (Fox & Bowman, 1966), but anteriorly the head is shortened and does not extend past the contact with the deltopectoral crest. A broad tuberosity located on the posterior head of the humerus would have supported the latissimus dorsi. Distally, the humerus is broadened and rectangular in shape, as described in Milleretta and Saurodektes (Gow, 1972; Reisz & Scott, 2002). The entepicondylar foramen is close to the posterior edge of the humerus, with well-defined proximal edges and a sloping distal surface as in Milleretta (Gow, 1972), although the entepicondylar foramen is positioned further proximally towards the entepicondyle anteriorly. A pronounced ectepicondyle and supinator process are present, as well as an ectepicondylar foramen. This ectepicondylar foramen originates in the groove between the ectepicondyle and supinator process, extends distally through the thickness of the ectepicondyle and opens next to the capitulum. The articulating surface of the humerus also resembles C. aguti, but the trochlea is shallower and does not extend as far proximally (Holmes, 1977). In addition, the capitulum is circular as opposed to the proximodistally oriented, oval-shaped capitulum in C. aguti (Holmes, 1977).

Figure 13 Left humerus of OMNH 73515.

(A) Proximal dorsal, (B) proximal ventral, (C) distal dorsal, (D) distal ventral, (E) proximal end and (F) distal views. Abbreviations: cap, capitulum; dpc, deltopectoral crest; ectp, ectepicondyle; ectp. for., ectepicondylar foramen; entp, entepicondyle; entp. for., entepicondylar foramen; l. dor. att., attachment point of latissimus dorsi; sup. pr., supinator process; tro, trochlea.

Discussion

Phylogenetic analysis

The purpose of our phylogenetic analysis is to test the position of Delorhynchus among parareptiles using the new anatomical data gleaned from this redescription of the holotype. Information provided by this study greatly enhanced the available cranial data for this pivotal early parareptile, but more importantly also allowed us to add valuable new postcranial data. We therefore updated the latest data matrix (Cisneros et al., 2021) with the new information about Delorhynchus, but restricted the analysis to a smaller number of taxa. We felt that there was no need to resort to using the full complement of Paleozoic amniotes and stem amniotes that was previously used and decided to test the relationships of the parareptile Delorhynchus within a smaller framework of parareptile evolution and see how our results compare to those of Cisneros et al. (2021). We carefully reevaluated all characters used by previous workers and made some modifications and corrections. All characters were resorted by anatomical position in a format similar to Laurin & Reisz (1995). Three characters were deleted from the analysis and six new characters were added for a total of 193 characters. Deleted characters include the presence of dorsal dermal ossifications, number of cusps of marginal dentition and contribution of the squamosal to the lower temporal bar, as these characters were not relevant to the taxa being studied due to inapplicability. Characters introduced in this study include an anterolateral shelf of the dorsal process of the maxilla, number of lateral exposures of the lacrimal, texture of the posterior edge of jugal, presence of the quadratojugal dorsal process under the skull roof, stapedial shaft morphology and the contact between the stapedial shaft and the stapedial dorsal process. In this process we also removed some parareptile taxa like Bashkyroleter and Rhipaeosaurus because their taxonomic identity is somewhat problematic due to consistently paraphyletic results in past studies and incompleteness of material respectively, and because their position among the ‘nycteroleters’ is unresolved (Tsuji, Müller & Reisz, 2012). In addition, we also removed the so-called ‘parareptile’ Carbonodraco from this analysis because we are not convinced that it is a parareptile, as reported in Mann et al. (2019), and we believe that its limited remains require further study. While several recent studies have provided alternate hypotheses into the relationships of Parareptilia (Ford & Benson, 2020; Cisneros et al., 2021; Simões et al., 2022), our analysis does not utilize the data matrices and results of two of these analyses given their broad scale and differing perspectives and, most importantly, the relatively small number of parareptilian taxa. While Ford & Benson (2020) places the 14 coded members of Parareptilia within Synapsida, Parareptilia is still present as a monophyletic clade of early amniotes. Simões et al. (2022) includes only 13 parareptiles in their analysis, and the resulting hypothesis in contrasts renders Parareptilia as paraphyletic. This particularly interesting study is very broad but uses too few parareptilian representatives, and excludes the oldest known well-preserved parareptile (Modesto et al., 2015). In strong contrast our primary goal with this phylogenetic analysis is to examine the relationships of Delorhynchus to all known parareptiles of western Pangaea more broadly, and given their relative comtemporaneity, test specifically its relationships to the multitude of parareptiles present at Richards Spur.

Results of analysis

The strict consensus of two most parsimonious cladograms of parareptile relationships is shown in Fig. 14. The tree length is 878 steps and has a consistency index (CI) of 0.2528, which is reduced to 0.2425 without uninformative characters and 0.0758 as a rescaled consistency index (RC). Including uninformative characters, the homoplasy index (HI) is 0.7472, and excluding uninformative characters it is 0.7575. Values for the retention index, f-value and f-ratio are 0.2999, 7605 and 0.3245 respectively. The data indicates that this is a weakly supported tree topology, but at the same time the results of our analysis confirm the status of Delorhynchus as a sister taxon to Colobomycter, within a clade that includes the known acleistorhinids. However the overall composition of the ‘acleistorhinids’ has a somewhat novel topology, with the larger clade that includes Feeserpeton, Lanthanosuchus and Acleistorhinus being successive outgroups to the clade comprising Karutia, Delorhynchus and Colobomycter. Perhaps most surprisingly, this clade of ‘acleistorhinids’ is the sister taxon to the other major clade of early parareptiles, the bolosaurids. This pattern is contrary to previous analyses which have placed bolosaurids in a more basal position among parareptiles, between the Gondwanan Milleretta and Australothyris (Cisneros et al., 2021). The interesting position of the Gondwanan Milleretta and Australothyris at the base of parareptiles also recovered previously (Cisneros et al., 2021) raises some interesting questions regarding the geographic origin and patterns of dispersal of parareptiles, especially in view of the undisputed fact that the oldest parareptiles are restricted to Laurasia.

Figure 14 Strict consensus of two most parsimonious cladograms of parareptile relationships (length 878 steps).

Bootstrap values (>50%) have been provided before the nodes.

There are some other notable differences between our results and those of previous studies. Interestingly, and not entirely unexpectedly, Mesosaurus becomes a sister taxon to the clade normally considered to be Eureptilia. The position of Mesosaurus among either parareptiles (Modesto, 2006; Cisneros et al., 2021) or among eureptiles (Modesto et al., 2015; Laurin & Piñeiro, 2017) is weakly supported and we suggest that this is because much of the anatomical evidence that could help evaluate the phylogenetic relationships of this taxon has been ‘overprinted’ by numerous autapomorphies.

The anatomy of the mysterious Lanthanosuchus and its closest relatives has always been somewhat controversial, again a taxon possessing a large number of autapomorphies and little-known postcranial anatomy. The most recent parareptile phylogeny (Cisneros et al., 2021) posited that Lanthanosuchus is a sister taxon to the ‘nycteroleters’. Instead, our results place this taxon as a sister to Acleistorhinus, as previously suggested (MacDougall et al., 2017b). We use the term ‘acleistorhinids’ in an informal manner because the clade now includes not only the usual suspects as shown in the most recent phylogenetic analysis (Cisneros et al., 2021), but also includes Lanthanosuchus.

Bootstrap values indicate that the position of Mesosaurus as a eureptile is weak, and clearly needs reevaluation. For example, the evidence for the presence of a lateral temporal fenestra is controversial (Modesto, 2006; Laurin & Piñeiro, 2017), as is the homology of the “swelling” of the vertebral neural arches. Furthermore, critical aspects of the braincase of the highly autapomorphic mesosaurs are missing, making their overall detailed anatomical reexamination worthwhile. Among ‘acleistorhinids’, only the bootstrap values for the clade of Delorhynchus and Colobomycter is high. We interpret the low values for the overall clade on a combination of factors, one being the very fragmentary nature of Karutia and the other being the lack of data on the postcranial anatomy of Lanthanosuchus and Acleistorhinus.

Conclusions

These results indicate that much work still needs to be undertaken for a better understanding of the initial stages of parareptile evolution, and hopefully new, more completely preserved fossil material like those described in this study will be uncovered. We refrained from considering the relationships of parareptiles with other amniote clades, as such a task is well beyond the objectives of this study. However, this new data together with emerging new information on other parareptiles and synapsids will provide the impetus for re-evaluating the position of ‘acleistorhinids’ and other parareptiles among amniotes.

Supplemental Information

Supplemental Information 1 Data matrix used for phylogenetic analysis.

Click here for additional data file.

Supplemental Information 2 Delorhynchus cifelli Full Character List for Data Matrix.

Each character for the phylogeny in this study, and the indication of changes where necessary as well as the source of each character where necessary.

Click here for additional data file.

We would like to thank Mr. W. J. May for his many years of generous contribution of Dolese material, and the staff of the Oklahoma Museum of Natural History for the loan of this specimen. We would also like to thank Ms. D. M. Scott in preparation and general assistance with anatomy.

Additional Information and Declarations

Competing Interests

Author Contributions

Data Availability

The authors declare that they have no competing interests.

Dylan C. T. Rowe analyzed the data, prepared figures and/or tables, authored or reviewed drafts of the article, and approved the final draft.

Joseph J. Bevitt performed the experiments, authored or reviewed drafts of the article, and approved the final draft.

Robert R. Reisz conceived and designed the experiments, authored or reviewed drafts of the article, and approved the final draft.

The following information was supplied regarding data availability:

The data is available at MorphoSource:

Media 000542097: Ct Data Of Complete Specimen [CTImageSeries] [NCT], https://doi.org/10.17602/M2/M542097.

Media 000542109: Segmentation Of Cranium And Associated Postcrania [Mesh] [NCT], https://doi.org/10.17602/M2/M542109.

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
