# Peer review of "Skeletal anatomy of the early Permian parareptile Delorhynchus with new information provided by neutron tomography"

_PeerJ, doi:10.7717/peerj.15935_

## Round 0.1 · original submission · Major Revisions

Dear Dr. Rowe,

I have now received the three reviews of your paper. As you will see, the reviewers provided detailed comments, and I trust these will be very useful when revising your paper. The reviewers highlighted-- inter alia-- various methodological aspects for you to clarify and address.

Apart from the comments of the reviewers, no mention is made of Carbonodraco lundi from Linton Ohio, and it seems reasonable to consider this taxon in the paper.

I am looking forward to your revised paper.

Sincerely,
Shaw

·

Basic reporting

Generally excellent. I suggest that a couple/few skeletal elements deserve their own figures. Authors “violate” ICZN Article 51 numerous times and this must be fixed. Their Materials & Methods section needs to be augmented. One reference is missing from the bibliography.

Experimental design

Excellent for the most part. Their Materials & Methods section needs to be augmented.

Validity of the findings

Excellent. I agree with their interpretations for the most part (exceptions noted in 4. Additional comments) and I was able to duplicate their phylogenetic results using their data matrix.

Additional comments

A nice, well-written manuscript that deserves to be published in the journal. There are numerous minor issues with formatting and phraseology, as noted below. I have suggested corrections not only for scientific accuracy but also because written English should be more “formal” than spoken English, e.g. there is no bone known as “the maxilla and nasal” but they are two distinct bones and it should appear as “the maxilla and the nasal”. Also, this creeping appearance of “anterior BONE” and “posterior BONE” phrasing (okay for a talk) when “anterior part of BONE” and “posterior part of BONE” is meant.

A concise summary of the phylogenetic methodology needs to be inserted into the Materials & Methods section (currently lines 80-91) Things to consider: was PAUP or TNT used? What search programs were used? Was a bootstrap analysis conducted and, If so, how many iterations? Were Bremer decal values for clades calculated and, if not, why not? All this is required for independent testing/replication. The link for the data matrix should be included here unless PeerJ requires it in its own section (the Morphobank link does not appear in the text).

Figures general: The leader lines should be thinner because in their present thickness they obscure some details in the figures.

Line 28: change “Delorhynchus is supported as a sister taxon of Colobomycter . . .” to “Delorhynchus is recovered as the sister taxon of Colobomycter . . . ” [note that each species or clade has but a single sister, so definite article required]

Lines 32-34: Concerning “The reptilian clade known as Parareptilia was first erected by Olson (1947) as one of
two subclasses of reptiles. It has been previously defined as the most inclusive clade containing Milleretta rubidgei and Procolophon trigoniceps (Tsuji & Muller, 2009).” [1] Olson (1947) does not appear in the references section. [2] the second sentence is too passive and there are other phylogenetic definitions for this clade; do you mean that you adhere to Tsuji & Muller (2009) definition and, if so, why? [3] the format “Procolophon trigoniceps (Tsuji & Muller, 2009)” implies that Tsuji & Muller (2009) originally erected the species in another genus; this is certainly not the case, so a little reformatting is necessary, perhaps “the clade Parareptilia sensu Tsuji & Muller (2009), who defined it as the most inclusive clade containing Milleretta rubidgei and Procolophon trigoniceps”.

Line 61: change “Macdougall” to “MacDougall”

Lines 121-123: concerning “This foramen extends posteromedially into the maxilla and is exposed on its medial surface at the middle of the dorsal lamina.” I cannot see this foramen in the medial view of the articulated maxilla (figure 3b). Because the maxilla is a character-rich element, it would be useful to have a figure of a “disarticulated” maxilla in several views.

Lines 142-143: with regards to “Dentition of the maxilla is largely homodont in shape and size . . . ” homodonty concerns shape and isodonty concerns size (cf. heterodonty and anisodonty, respectively).

Line 165: the format “Acleistorhinus pteroticus (deBraga & Reisz, 1996)” implies that deBraga & Reisz (1996) originally erected the species in another genus; this is certainly not the case, so please reformat. See also lines 195-196, line 304, and elsewhere.

Line 172: change “. . . Another smooth suture between the nasal and prefrontal” to “. . . Another smooth suture between the nasal and the prefrontal” [there is no such bone as “ the nasal and prefrontal”]

Line 179: change “while” to “whereas”. Do the same for lines 482, 491, 541, 825.

Line 167: concerning “MacDougall et al., 2017b” there are two MacDougall et al., 2017 references listed in lines 1038 to 1045 but neither is coded “a” or “b”.

Line 182: I am a not sure what “. . . the anterior part of the narial opening” means; it suggests to me that the septomaxilla is positioned anteriorly in the external naris and this is surely not the case.

Lines 184-186: with regards to “It is unclear, based on the extent of preservation, whether a septomaxillary foramen would be present as described in C. pholeter (MacDougall et al., 2017b).” Yes, the septomaxillary foramen is unclear in MacDougall et al. (2017b) because they did not label it in their illustrations, but it can be seen clearly in their figure 4b. Based on this observation, I would interpret a septomaxillary foramen of similar morphology in D. cifellii.

Line 189: “exposure between the maxilla and THE nasal”

Lines 189-191: comment on “This unusual feature, with a significant portion of the bone being covered on the external surface of the snout, is also present in Colobomycter pholeter”—this unusual feature was first described in Colobomycter pholeter by MacDougall et al. (2017), i.e. not just “also present”. Also, both lateral exposures should be labelled in figure 2b; the submission version of the figure places a leader line over the anterior lateral exposure of the lacrimal, partially covering it and thereby obscuring it.

Line 194: with regards to “the anterior prefrontal” I know that this kinf of phraseology is a verbal shortcut we use when talking shop, but it should be avoided (it suggests that there is also a posterior prefrontal, and further that there are two pairs for prefrontals) and the correct “the anterior part of the prefrontal” should be used in a formal, scientific description. The same idea pertains to “posterior lacrimal” that appears in lines 212-213.

Lines 208-209: change “In Colobomycter vaughni (MacDougall, Modesto & Reisz, 2016), the lacrimal duct is . . .” to “According to MacDougall, Modesto & Reisz (2016), the lacrimal duct of Colobomycter vaughni is . . .” to avoid the formatting problem noted for line 165 and elsewhere.

Lines 210-211: concerning “the lacrimal duct is enclosed by the maxilla laterally and soft tissue medially” there is no preserved soft tissue so this phrase needs to be reworked.

Lines 227-228: with regards to “The posterior process of the jugal does not contact the squamosal in this specimen, which matches what has been suggested in Haridy et al. (2016)”—this is true as preserved, but if one were to reposition the posterior part of the temporal skull roof (the squamosal and the quadratojugal) with the anterior part (the postorbital and the jugal) I can envision the jugal making contact with the squamosal using figure 2b.

Line 243: concerning the descriptive “the basal bolosaurid Belebey vegrandis”: it is not basal according to your own tree topology!

Line 339: “contacts the postorbital and THE supratemporal medially”

Lines 360-362: change “In contrast, the occipital flange that contributes to the quadrate foramen, as well as the process supporting the lateral condyle of the quadrate, is smooth on the external surfaces” to “In contrast, the external surface of occipital flange that contributes to the quadrate foramen, as well as the process supporting the lateral condyle of the quadrate, is smooth.”

Lines 354, 358: change “while” to “whereas”

Lines 338–351: does the squamosal contribute to the quadrate foramen at all? Superficially?

Line 376: change “Macroleter & Owenetta” to “Macroleter and Owenetta”

Line 380: the parasphenoid is not considered part of the palate proper. Also, change “the vomer, palatine, pterygoid, ectopterygoid” to “the vomer, the palatine, the pterygoid, the ectopterygoid”.

Line 389: change “through the ectopterygoid and pterygoid” to “across the ventral surfaces of the ectopterygoid and the pterygoid”

Line 401: I am not so sure that the vomer largely defines “the shape of the snout” and this is not at all clear from the figures.

Line 416: change “much of the lateral palate” to “much of the lateral part of the palate”

Line 417: change “consists of the palatine (Fig. 4). It is bordered” to “consists of the palatine (Fig. 4). This element is bordered”

Line 420: the suborbital foramen does not look pronounced on the (anatomical) left side.

Line 431: The ectopterygoid is a relatively large element OF the palate

Line 462-464: looks like there is a tympanic flange coming off the quadrate ramus in figure 4.

Line 472: italicize “Macroleter poezicus” and move the citation elsewhere as per my comments for line 165.

Line 474: change “articulation, while posteriorly it overlies the anteromedial quadrate ramus of the .” to “articulation, whereas posteriorly it overlies the anteromedial part of the quadrate ramus of the ptertgoid.”

Line 476: “the epipterygoid . . . posteromedially and approached or possibly connected to the skull roof.” You would have a better idea if you produced a skull reconstruction.

Lines 478–500: the description of the parabasisphenoid should be part of the braincase section. I have to remark that the dark blue used to colour the parabasisphenoid in figure 4 washes out and obscures the detail on this element! Can it be changed to medium gray or equivalent?

Line 486: with regards to the phrase “overlies the medial gap between the pterygoids”—this “gap” is the interpterygoid vacuity and it should be referred to as such. Also, does the cultriform process bisect the interpterygoid vacuity fully? Or partially?

Line 489: the species Lanthanosuchus watsoni was not erected by deBraga & Reisz (1996).

Lines 495–497: “The basipterygoid processes extend anterolaterally and would suture onto the pterygoid between the anterior process and quadrate ramus in a fully articulated specimen.” I don’t see any evidence in figure 4 that the basipterygoid processes would not form synovial joints with the basicranial recesses, so why infer sutures?

Lines 497–499: concerning “As in Macroleter poezicus (Tsuji, 2006), the sella turcica is positioned caudally to the dorsal body, bordered by the clinoid processes posterolaterally. While the clinoid processes are . . .”—I am not familiar with the term “dorsal body” with regards to parabasisphenoid morphology. Furthermore, the parabasisphenoid, like the maxilla, is a character-rich element and deserves a separate figure inasmuch the authors can then label features such as the sella turcica and clinoid processes.

Lines 499–501: change “While the clinoid processes are smooth on their medial and lateral surfaces, the posterodorsal surface has a foramen which would envelop cranial nerve VI” to “The medial and lateral surfaces of each clinoid process is smooth, and the foramen for cranial nerve VI pierces each process posterodorsally.”

Lines 509–511: In “As in Colobomycter pholeter (MacDougall et al., 2017b), the contact with the articular facet is secure, with protuberances of the quadrate fitting into the dorsal sockets of the articular” I am not quite sure what is going on here. Is the articular facet here on the quadrate condyles? Are the protuberances the quadrate condyles? Are the the dorsal sockets of the articular the cotyles of the articular?

Lines 518-521: A couple of comments on the passage “This element is shifted out of place, with the ventral margin partially contacting the cultriform process of the parabasisphenoid (Fig. 4A). In a fully articulated specimen, the ventral margin of the sphenethmoid would likely contact the trough of the cultriform process.” [1] the contact between the sphenethmoid and the cultriform process of the parabasisphenoid cannot be seen clearly in the figures provided (more argument for a separate figure for the latter bone) and [2] in figure 4a it looks like the sphenethmoid has been rotated about 90 degrees, but I am not sure because a clear view of this element in lateral view is not presented. Despite these comments, I do agree that, in life, the sphenethmoid sat in the trough of the cultriform process.

Line 526: “Delorhynchus cifellii” needs to be italicized

Line 533: the phrase “This element is massively built, in contrast to captorhinids such as. . .” is strange to me because, having researched captorhinids, I think that captorhinid stapes are “massively built”. Looking at figure 4, the stapes looks like a compact, robust element, but I would, myself, not describe it as “massively built”. For reference, the stapes in figure 4 does not appear to be much larger than the condylar portion of the quadrate. In specimens of Captorhinus, the stapes is huge with respect to the condylar portion of the quadrate; or put another way, the length of the stapes in Captorhinus is about 1/4 total skull length. The same cannot be said for the stapes of Delorhynchus.

Line 568: delete the word “ventral”

Lines 568-569: change “This posteroventral braincase element” to The basioccipital”

Lines 574-575: with “The shape of the paired exoccipitals therefore more closely resembles what is seen in Owenetta kitchingorum (Reisz & Scott, 2002)” you must mean “The shape of the paired exoccipitals therefore more closely resembles what is reconstructed for Owenetta kitchingorum by Reisz & Scott (2002)” because the exoccipitals in their specimen drawings look like blobs (sorry).

Line 576: “The neurocranial complex” or neurocranium is comprised of the basisphenoid, the supraoccipital, the prootics, the opisthotics, and the ex- and basioccipitals. The supraoccipital-opisthotic (and prootic?) complex is another structure that needs its own figure to document it properly (authors provide only “ghost” views in figure 6). This would help to mitigate misinterpretations of, e.g., supraoccipital morphology, by researchers who are not intimately familiar with parareptile morphology, particularly those colleagues of ours who research anamniotes but are scoring amniotes for a broader phylogenetic analysis of tetrapods.

Line 591: “venous notches”—please label!

Lines 625–230: so few hyoid elements have been described for parareptiles that I would hope that the authors compare those here with those of, e.g. Owenetta kitchingorum as documented by Reisz and Scott (2002).

Line 641: what is “a dorsoventral break” ? Is this in the frontal plane of the specimen? Not clear to me.

Lines 642–644: regarding “The left section of the mandible, as a result, is rotated such that the dentary lies almost perpendicular to the maxilla”—the left dentary does not look rotated with respect to the skull or to the other dentary/mandicular ramus. Please clarify.

Line 646: with regards to “The posterolabial surface of the dentary.. .” I think the authors should restrict aspect terms such as “labial” to teeth and so rephrase as ““The posteromedial surface of the dentary.. .” Same applies to line 648, 652, 653 (lingual), 703 (lingual), 716 (lingual)

Lines 651–652: change “the lateral surangular” to “the lateral surface of the surangular”

Line 655: in what way is the Meckelian canal “C-shaped” ? Not clear to me.

Line 656-657: “a deep foramen branches off from the Meckelian canal. This foramen. . .”—I cannot see such a foramen in figure 7. Please label, and identify if possible.

Line 702: change “the posterior mandible” to “the posterior part of the mandible”

Lines 702-707: given that OMNH 73515 is a smaller individual than OMNH 72363, would it be fair to say that the prearticular makes contact with the anterior coronoid as this species gets larger? (older?)

Line 707: orphan word (“It”) at end of line.

Lines 709–710: concerning the sentence “Compared to the thoroughly described articular in Captorhinus laticeps, this element is much shorter (Heaton, 1979)”—be aware that Heaton used Richards Spur captorhinid specimens (probably referable to C. aguti) to supplement his description and illustrations of the cranial anatomy of “Eocaptorhinus” laticeps, so his reconstruction of the articular is a mix of information from at least two captorhinid species. This is my long-winded way of stating that one is better off using Fox & Bowman (1966) or Modesto (1998) or both for information on the articular of a species of Captorhinus.

Line 719: change “nearly full complement of the atlas-axis complex is present” to “nearly complete atlas-axis complex is present”

Line 733-734: change “Dorsally, the atlantal centrum is flattened where it would support the spinal cord” to “Dorsally, the atlantal centrum is flattened where contributes to the vertebral foramen.”

Lines 752-753: change “such as Captorhinus aguti (Fox & Bowman, 1966) and Eocaptorhinus laticeps (Dilkes & Reisz, 1986)” to “such as Captorhinus aguti and Eocaptorhinus laticeps (Fox & Bowman, 1966; Dilkes & Reisz, 1986)”

Line 756: change “C. aguti (Fox & Bowman, 1966)” to “C. aguti (figure 36 in Fox & Bowman, 1966)”

Lines 776-777: concerning “C. aguti (Fox & Bowman, 1966)” refer the reader to a specific figure in Fox & Bowman (1966).

Lines 791-798: the directional terms “distally”, “proximal”, and “distal” appear in the description of the ribs. These terms are directional terms traditionally applied to limbs and their structures because “proximal” refers to the attachment point of the limb and “distal” refers to the opposite end, given that a limb’s position can very with respect to the head and trunk (particularly in human anatomy). It would be preferable to use “dorsal” and “ventral” when referring to the ends of ribs because these structures are part of the trunk (i.e. not limbs).

Line 826: what is the “posterior scapula” ?

Lines 857 -858: concerning “The proximal head of the humerus is porous”—are you referring to the area that formerly bore the joint cartilage? Or are you describing unfinished bone?

Line 865: change “further up” to “further proximally”

Line 902: change “since” to “because”

Lines 904–906: change “we also removed the so-called ‘parareptile’ Carbonodraco (Mann et al., 2019) from this analysis because we are convinced that its taxonomic identity and indeed its most recent anatomical interpretation are highly problematic” to “we also removed Carbonodraco from this analysis because we are not convinced that it is a parareptile, as reported by Mann et al. (2019), and we believe that its limited remains require further study”.

Lines 923-925: comments on “this clade of ‘acleistorhinids’ are the sister taxon to the other major clade of Permo-Carboniferous parareptiles, the bolosaurids”—[1] grammatically this passage should begin “this clade of ‘acleistorhinids’ is the sister taxon. . .” and [2] I disagree that Bolosauria is the ‘other major clade of . . . parareptiles”—that would be the smallest clade that includes Nyctiphruretus and Bradysaurus in figure 11 (= Procolophonia?).

Figure 4: concerning the abbreviation “nc”, the structure labelled is not the entire neurocranium (the basisphenoid is fused to the parasphenoid) and appears to be coalescled supraoccipital and opisphotics, so I would prefer the label to reflect this (as, perhaps “so + op + pro”)

Figure 7: the two coronoids should be coloured differently. This figure needs a lateral view of a mandibular ramus to show an unobstructed view of the dentition et al.

Sean Modesto,
Sydney, Canada
23 February 2023

Reviewer 2 ·

Basic reporting

The manuscript provides valuable data on one of the most complete specimens of an early reptile known to date and which might be important to understand some aspects of early reptile relationships. This is based on high resolution neutron CT scans, and detailed segmentation of all components of the skeleton. For this I commend the authors for their work. However, there are important areas of the manuscript that could benefit from more detailed information in the description, discussions based on the most recent literature in the topic, besides issues with data availability that I have highlighted below in my comments.

Introduction: in the very first sentences of the introduction the authors discuss parareptiles as a clade with some older references about this group. They completely ignore more recent studies that have used much larger datasets and new analytical techniques that have found parareptiles as paraphyletic and/or in new placements in the early amniote tree of life compared to previous hypotheses (Ford & Benson 2020; Simões et al. 2022). I suggest including these considerations to make justice to the recent literature.

Anatomical description:
1) The authors have extremely long paragraphs that sometimes combine the description for different bones (e.g., premaxilla and maxilla). Also, there is no paragraph indents nor line spacing between paragraphs, making the text a bit difficult to read. I would suggest authors reviewing this formatting aspects and adjusting accordingly.

2) Differences between the original description of this holotype (Reisz et al. 2014) and this study are not stated. However, there are important differences between the original description and the new anatomical interpretations here. One important example is in the temporal region. In the original description of this specimen the squamosal was much smaller and the supratemporal bone was interpreted as forming most of the posterolateral corner of the skull roof. However, the CT scan data interpretation by the authors here suggest the squamosal is much larger than previously described and forms that area of the skull, with the supratemporal being much smaller and having a very different shape than previously described (Fig. 2 here). I strongly suggest the authors to discuss this and other examples where differences exist in the relevant areas of the text.

Experimental design

Materials and Methods: There is nearly no description of the phylogenetic methods used in this study. Their entire description of this portion of their study consists only of the following sentence: “The phylogenetic analysis in this study used PAUP 4.0a149, and the matrix from Cisneros et al. (2020) was updated in Mesquite.” This is unacceptable and the authors must describe what changes they have made to this dataset, the search algorithms used for phylogenetic tree inference, how they calculated support for each node, etc. Some of these were mentioned briefly in the phylogenetic results section (lines 881-914), but not in enough detail besides the fact they are in the incorrect section of the manuscript.

Validity of the findings

Data accessibility and reproducibility: during the review process I had difficulties accessing the CT scan files associated with this study. After several interactions with the editors and authors, I can now have access to the media files with the instructions provided by the authors by email. However, there are still important problems associated with data availability:

1) Nowhere in the manuscript there are instructions to access those files in the repository (which is why I had no idea where to find them initially);

2) Morphobank is not a repository for this type of data (this is why all files had to be bundled together and uploaded as a zipped file). Most readers would never assume that CT scan files are deposited there and most likely will never see it. CT scan files should be made available through appropriate repositories that can handle and curate CT scan data, such as Morphosource;

3) The author said: "However, our lab has had issues launching Avizo projects when the file locations have changed, and therefore the reviewer may not be able to access the segmentation files.". This should not be an issue if they actually export 3D volume meshes (e.g., .ply or .stl format, not Avizo formatted files). Volume files can be opened anywhere at any time by anyone. This is really foundation knowledge in handling CT scan files. I appreciate the authors have made the raw data available, but volume meshes are necessary to be made available too as they actually reflect the authors’ interpretation of the data during the segmentation process and subsequent anatomical descriptions. It will also be to the authors own benefit as other researchers will be able to use their results and cite this study are frequently.

Additional comments

Lines 51-53: “The preservation of such a high taxic diversity of parareptiles suggests that this region was a center of parareptilian diversification in the early Permian, especially for acleistorhinids (MacDougall, Modesto & Reisz, 2016).” Preservation of a large number of fossil species does not necessarily correlate with actual species richness. We have a vast literature on the biases in the fossil record showing this, and several tools are available precisely to correct for taphonomic (preservation) biases in studies of lineage diversification in the fossil record, such as SQS sampling to name just one. In this particular instance, this locality is an accretion deposit that has accumulated several individuals across the span of several hundreds of thousands of years, if not longer. Specimen abundance and species richness reflect a long-term compilation of individuals that did not live at the same time and perhaps were not even inhabiting the same locality, as a lot of these materials are allochthonous. So the authors definitely cannot state the sentence above based on the data they have available.

Fig. 3: there is a very robust medial process on the medial surface of the jugal that is unlike that seen in any other reptile that I know of. This is not described or mentioned in the description. May the authors clarify what this process is?
Lines: 893-896: What were the criteria to remove characters from the dataset? These should be listed clearly.

Lines: 897-901: You must list new characters appropriately. The current way those new characters are described do not tell what the character number is in the character list, do not tell the character states for each character, and basically do not follow any standard guidelines for creating new characters. See standard character format guidelines in Sereno (2007).

Lines 903-907: “…we are convinced that its taxonomic identity and indeed its
906 most recent anatomical interpretation are highly problematic.” . This sentence is confusing. Please re-phrase.

Lines 901-907: What were the criteria to consider the taxonomic identity of Bashkyroleter and Rhipaeosaurus as ”problematic”? This is a vague statement that does not actually address the reasons for removing those species nor what is the logic behind their criticism for the taxonomic identity of these species.

Lines 932-939: estimation for the introduction, this is an updated discussion as it does not include the results from the most recent and largest available phylogenies exploring the relationship of parareptiles among early amniotes (Ford & Benson 2020; Simões et al. 2022). The relationships within parareptiles are only relevant to discuss if this is a monophyletic group to begin with, which might well not be the case. This should be all discussed in the text to make the discussion up-to-date and more interesting to the reader.

Lines 950-951: “For example, the evidence for the presence of a lateral temporal fenestra is controversial (Modesto, 2006; Laurin & Piñeiro, 2017)”. I’m not sure if this is actually controversial. The most recent study on this topic (the 2017 paper listed) is in fact quite convincing and there has been no subsequent papers to that refuting your interpretations or showing better preserved specimens without a temporal fenestration. To see that this is controversial there must be an actual debate in the literature or evidence presented here.

Lines 964-969: The conclusion is very brief and once again ignores the most recent studies (Ford & Benson 2020; Simões et al. 2022). See more comments above.

References
Ford, D. P. & Benson, R. B. 2020. The phylogeny of early amniotes and the affinities of Parareptilia and Varanopidae. Nature Ecology & Evolution, 4(1), 57-65.
Reisz, R. R., Macdougall, M. J. & Modesto, S. P. 2014. A new species of the parareptile genus Delorhynchus, based on articulated skeletal remains from Richards Spur, Lower Permian of Oklahoma. Journal of Vertebrate Paleontology, 34(5), 1033-1043.
Sereno, P. C. 2007. Logical basis for morphological characters in phylogenetics. Cladistics, 23(6), 565-587.
Simões, T. R., Kammerer, C. F., Caldwell, M. W. & Pierce, S. E. 2022. Successive climate crises in the deep past drove the early evolution and radiation of reptiles. Science advances, 8(33), eabq1898.

·

Basic reporting

The manuscript presents an updated description of the holotype cranium and postcranium of the acleistorhinid reptile Delorhynchus. It is based on high quality neutron CT scans and it reveals new anatomical information. The figures are of excellent quality and the literature cited is up to date. There are only a few typos and some omissions (see below).

Experimental design

The methods used for the acquisition and treatment of the CT-images are succinctly described. The descriptive portion of the work is well done and supported by the images provided.

The methods used for the phylogenetic analysis, on the other hand, are not described in sufficient detail. We only know that some taxa and characters were deleted from the previous matrix, apparently some changes made to the character states, and the search was done in PAUP. However, we are not told what kind of search it was, how many trees per replicate were saved, if there were ordered or weighted characters, which collapsing rule was used, etc. On top of that, we are not informed of the new codings, which source of information was used for the new codings, and the character-taxon matrix was not provided, so we do not know which changes were actually made and why.

Validity of the findings

Although the anatomical description is solid and helpful, the absence of sufficient information on the methods section makes it nearly impossible the replicate and evaluate the phylogenetic section of this study. This is critical to a study that challenges previous results regarding the phylogenetic placement of some parareptiles.

The manuscript cannot be accepted in its current version. More methodological informations need to be presented. The datamatrix should be provided so we can replicate the results. New codings need to be explained and justified. Only then we can assess the validity of the findings.

Additional comments

Note that 'Owenetta' kitchingorum is a junior synonym of Saurodectes kitchingorum (sensu Hamley et al. 2021, Zoological Journal of the Linnean Society). The taxon 'Owenetta' is cited a few times in the text for comparison, but I am not sure if the authors are referring to the type species Owenetta rubidgei from the Permian or to the Triassic Saurodektes kitchingorum. In line 575, it is clear that they refer to the latter.

---

## Round 0.2 · Major Revisions

Dear Dr. Rowe,

I have now received the two reviews of your paper.

The one reviewer suggest some minor changes, and these will be easy to address.

The second reviewer pointed out several aspects for you to address. Importantly, the second reviewer pointed out the lack of open access to the data of the paper. This is a most crucial aspect for you to address in the spirit of open science.

I am looking forward to the revised paper.

sincerely,
Shaw

·

Basic reporting

Excellent.

Experimental design

Excellent.

Validity of the findings

Excellent.

Additional comments

This version is an improvement on the original version. A few corrections remain to be made:


Lines 45-48: formatting in the sentence “These include the basal parareptile Microleter mckinzieorum (Tsuji, Müller & Reisz, 2010), the bolosaurid Bolosaurus grandis (Reisz, Barkas & Scott, 2002), the nyctiphruretid Abyssomedon williamsi (MacDougall & Reisz, 2014), the lanthanosuchoid Feeserpeton oklahomensis (MacDougall & Reisz, 2012) .. .” imply [1] that that Tsuji, Müller & Reisz (2010) named Microleter mckinzieorum in another genus, [2] Reisz, Barkas & Scott (2002) named Bolosaurus grandis in another genus, [3] MacDougall & Reisz (2014) named Abyssomedon williamsi in another genus, and [4] MacDougall & Reisz (2012) named Abyssomedon williamsi in another genus, none of which is correct.

Line 60: in “D. cifellii which”, the word “which” should follow a comma (because the word “which” usually begins a nonrestrictive clause in English grammar).

Line 101-106: the sentence beginning “The tree length is 878 steps . . .” does not belong in the Materials & Methods section. Transfer it to the “Results of analysis” section (line 879).

Line 161: “Acleistorhinus pteroticus (Daly, 1969)” means Daly (1969) erected Acleistorhinus pteroticus in another genus originally. Change wording to, e.g. “Acleistorhinus pteroticus (figures 1-3 in Daly, 1969)”

Line 163: “C. pholeter (MacDougall et al., 2017b)” means MacDougall et al. (2017b) erected C. pholeter in another genus originally. Change wording to, e.g. “C. pholeter (figure 3d, e in MacDougall et al., 2017b)”. See also lines 181, 242-243, 305, 351-352, 411, 454, 480

Lines 190-191: change “Acleistorhinus pteroticus (deBraga & Reisz, 1996)” to follow ICZN Article 51. See also lines 253, 536, 700

Line 204: fix double comma

Line 244: change “Belebey vegrandis (Reisz et al., 2007)” to follow ICZN Article 51

Line 246: change “C. vaughni (MacDougall, Modesto & Reisz, 2016)” to follow ICZN Article 51. See also lines 265, 278

Line 396: insert space in “region(Fig.”

Line 468: change “Macroleter poezicus (Tsuji, 2006)” to follow ICZN Article 51. See also lines 494, 575

Line 487: change “Lanthanosuchus watsoni (deBraga & Reisz, 1996)” to follow ICZN Article 51. See also line 489

Line 497: fix double full stop

Line 543: change “Milleretta rubidgei (Gow, 1972)” to follow ICZN Article 51. See also line 549

Line 557: change “Mesenosaurus romeri (Reisz & Berman, 2001)” to follow ICZN Article 51.

Line 569: insert space in “Scott(2002)”

Line 586: change “Dimetrodon limbatus (Romer & Price, 1940)” to follow ICZN Article 51.

Line 628: change “Delorhynchus cifellii (Haridy, MacDougall & Reisz, 2018)” to follow ICZN Article 51.

Line 628: insert full stop in “2018) The”

Line 637: change “Karutia fortunata (Cisneros et al., 2020)” to follow ICZN Article 51.

Lines 637-638: change “and D. multidentatus (Rowe et al., 2021)” to follow ICZN Article 51.

Line 708: change “Captorhinus aguti (Fox & Bowman, 1966)” to follow ICZN Article 51. See also lines 757-758, 794, 825, 833

Line 873: insert space in “becausetheir”

Line 732: Eocaptorhinus has been regarded a junior synonym of Captorhinus for three decades now.

Line 1035: italicize “Belebey chengi”

Line 1047: italicize “Concordia”

Reviewer 2 ·

Basic reporting

Basic reporting
The manuscript provides valuable data on one of the most complete specimens of an early reptile known to date and which might be important to understand some aspects of early reptile relationships. This is based on high resolution neutron CT scans, and detailed segmentation of all components of the skeleton. For this I commend the authors for their work. However, there are important areas of the manuscript that could benefit from more detailed information in the description, discussions based on the most recent literature in the topic, besides issues with data availability that I have highlighted below in my comments.

Introduction: in the very first sentences of the introduction the authors discuss parareptiles as a clade with some older references about this group. They completely ignore more recent studies that have used much larger datasets and new analytical techniques that have found parareptiles as paraphyletic and/or in new placements in the early amniote tree of life compared to previous hypotheses (Ford & Benson 2020; Simões et al. 2022). I suggest including these considerations to make justice to the recent literature.

We have thoroughly analyzed the results of both papers, and while Ford & Benson (2020) places Parareptilia as a sister group to synapsids the clade is still recognized and therefore does not impact our study. In Simões et al. (2022) parareptiles do not come out as a clade but crucial taxa such as bolosaurids and lanthanosuchoids are ignored, and we therefore prefer to stay conservative and recognize the clade of Parareptilia.

Reviewer: I find this response insufficient. Of the two clades not included in the analysis of Simoes et al. (2022) mentioned by the authors, lanthanosuchoids is historically the same as acleistorhinids + Lanthanosuchus. Acleistorhinids were included, and so does the bulk of lanthanosuchoids. Also, the taxonomic identity and systematics of Lanthanosuchus is highly debatable and the material is not currently available for study, so this criticism is not valid. But the crucial point here is that all dataset can still be expanded. Yet, this is the largest one we have available, and so not considering its results due to sample size/coverage and instead cherry picking older results based on a substantially smaller samples (not even health the size of the two studies mentioned above), is ludicrous. I find it extremely problematic from a scholarly perspective this preferential selection of results that are convenient to the authors in face of their former published hypotheses rather than the latest advances in the field.

Anatomical description:

2) Differences between the original description of this holotype (Reisz et al. 2014) and this study are not stated. However, there are important differences between the original description and the new anatomical interpretations here. One important example is in the temporal region. In the original description of this specimen the squamosal was much smaller and the supratemporal bone was interpreted as forming most of the posterolateral corner of the skull roof. However, the CT scan data interpretation by the authors here suggest the squamosal is much larger than previously described and forms that area of the skull, with the supratemporal being much smaller and having a very different shape than previously described (Fig. 2 here). I strongly suggest the authors to discuss this and other examples where differences exist in the relevant areas of the text.
Regarding the squamosal, the size has not changed significantly between Reisz et al. (2014) and this study when comparing the squamosal previously illustrated in OMNH 73362 and the squamosal modelled in OMNH 73515. As mentioned in the text, the supratemporal was not fully preserved in this specimen.


Reviewer: I find this response insufficient. I fail to understand why the authors continue to ignore differences between the former description and the new results obtained from the CT scan in their manuscript. I strongly disagree with their interpretation that there is no difference between the former and the present anatomical data presented (e.g., the squamosal size). At best, other readers may have the same impression as I do and this should be discussed. Otherwise, what was the point of the CT scan if nothing new is gained in terms of anatomical interpretation?

Experimental design

Experimental design
Materials and Methods: There is nearly no description of the phylogenetic methods used in this study. Their entire description of this portion of their study consists only of the following sentence: “The phylogenetic analysis in this study used PAUP 4.0a149, and the matrix from Cisneros et al. (2020) was updated in Mesquite.” This is unacceptable and the authors must describe what changes they have made to this dataset, the search algorithms used for phylogenetic tree inference, how they calculated support for each node, etc. Some of these were mentioned briefly in the phylogenetic results section (lines 881-914), but not in enough detail besides the fact they are in the incorrect section of the manuscript.

We have noted and incorporated these changes.

Reviewer: Good. Yet, the authors still did not include the tree search algorithm used in PAUP, and the parameters chosen for appropriate replication.

Validity of the findings

Validity of the findings
Data accessibility and reproducibility: during the review process I had difficulties accessing the CT scan files associated with this study. After several interactions with the editors and authors, I can now have access to the media files with the instructions provided by the authors by email. However, there are still important problems associated with data availability:

1) Nowhere in the manuscript there are instructions to access those files in the repository (which is why I had no idea where to find them initially);

2) Morphobank is not a repository for this type of data (this is why all files had to be bundled together and uploaded as a zipped file). Most readers would never assume that CT scan files are deposited there and most likely will never see it. CT scan files should be made available through appropriate repositories that can handle and curate CT scan data, such as Morphosource;

3) The author said: "However, our lab has had issues launching Avizo projects when the file locations have changed, and therefore the reviewer may not be able to access the segmentation files.". This should not be an issue if they actually export 3D volume meshes (e.g., .ply or .stl format, not Avizo formatted files). Volume files can be opened anywhere at any time by anyone. This is really foundation knowledge in handling CT scan files. I appreciate the authors have made the raw data available, but volume meshes are necessary to be made available too as they actually reflect the authors’ interpretation of the data during the segmentation process and subsequent anatomical descriptions. It will also be to the authors own benefit as other researchers will be able to use their results and cite this study are frequently.

We do not agree that we have to share further data as many other paleontological studies involving CT only share the raw data necessary for segmentation. We respectfully disagree with this reviewer, as Morphobank includes the option for uploading 3D media as quoted here: “Click here to load zipped archives of CT scan stacks (Dicom, TIFF) as well as PLY and STL type files or video files”. In addition, Morphobank is consistently open-access and available to everyone in contrast to MorphoSource, where access to project data must be requested.

Reviewer: Incorrect. Access to projects in in MorphoSource do not need to be closed. It is up to the researcher uploading the project to choose it, and so the authors here can promptly make it fully open without request if they wish. Also, even when a quick request is required for projects, it is still open (all you need to do is ask). As for MorphoBank, the media uploaded there is not curated in the same way as standard repositories for CT scan data. Finally, the authors continue to NOT provide the link to the data (even for MorphoBank).

As to the sentence “We do not agree that we have to share further data as many other paleontological studies involving CT only share the raw data necessary for segmentation”, I find it alarming that in this day and age, where open data access is not only necessary but also easily achievable to maintain scientific integrity and reproducibility, that some authors continue to abide to the practice of hiding their data. It is a basic logic fallacy that, if others have done it in the past, that the present authors are entitled to repeat this practice.

Additional comments

Additional comments
Lines 51-53: “The preservation of such a high taxic diversity of parareptiles suggests that this region was a center of parareptilian diversification in the early Permian, especially for acleistorhinids (MacDougall, Modesto & Reisz, 2016).” Preservation of a large number of fossil species does not necessarily correlate with actual species richness. We have a vast literature on the biases in the fossil record showing this, and several tools are available precisely to correct for taphonomic (preservation) biases in studies of lineage diversification in the fossil record, such as SQS sampling to name just one. In this particular instance, this locality is an accretion deposit that has accumulated several individuals across the span of several hundreds of thousands of years, if not longer. Specimen abundance and species richness reflect a long-term compilation of individuals that did not live at the same time and perhaps were not even inhabiting the same locality, as a lot of these materials are allochthonous. So the authors definitely cannot state the sentence above based on the data they have available.

We find that this comment is misdirected, as the Richards Spur locality is unique in the Paleozoic as it represents more species than all other early Permian localities combined in the form of pockets of fossiliferous material, which maintain high levels of taxic diversity. We have, however, reworded this sentence to be more conservative.

Reviewer: the authors seem to have not fully understood my statement. Accretion deposits with allochtonous materials cannot be used to infer ecological information (such as species richness), as the latter depends on a well detailed understanding of where species inhabited in time and space. To make it into a more plain language: a bunch of fossils accumulated in the pit that came from various geographical locations in different time periods telling nothing of the latter.


Lines: 893-896: What were the criteria to remove characters from the dataset? These should be listed clearly.

Error noted and corrected.

Reviewer: the authors corrected this by mentioning “as these characters were not relevant to the taxa being studied”. What do you mean by not being relevant? This does not really answer it. Do you mean they are inapplicable or invariant?

Lines: 897-901: You must list new characters appropriately. The current way those new characters are described do not tell what the character number is in the character list, do not tell the character states for each character, and basically do not follow any standard guidelines for creating new characters. See standard character format guidelines in Sereno (2007).

We find that this comment is misdirected, as a list of characters including character states and new characters was provided in the supplementary files.

Reviewer: thank you for adding the character list.


Lines 932-939: estimation for the introduction, this is an updated discussion as it does not include the results from the most recent and largest available phylogenies exploring the relationship of parareptiles among early amniotes (Ford & Benson 2020; Simões et al. 2022). The relationships within parareptiles are only relevant to discuss if this is a monophyletic group to begin with, which might well not be the case. This should be all discussed in the text to make the discussion up-to-date and more interesting to the reader.

As mentioned above, we do not wish to discuss these papers here as comments would be better made elsewhere.

Reviewer: see comments above.

---

## Round 0.3 · accepted · Accept

Dear Dr. Rowe,

Thank you for the revised manuscript. You have addressed all the comments of the last round of reviews, and I am pleased to inform you that your paper has now been accepted for publication.

Sincerely,
Shaw Badenhorst